# A Bayesian machine learning approach for drug target identification using diverse data types

Neel S. Madhukar [1,2,3,4,7,8], Prashant K. Khade[5,8], Linda Huang[1,2,3], Kaitlyn Gayvert [1,2,3,4], Giuseppe Galletti[5], Martin Stogniew[6], Joshua E. Allen[6]*, Paraskevi Giannakakou[3,5]* & Olivier Elemento[1,2,3,4,7]*

Drug target identification is a crucial step in development, yet is also among the most complex. To address this, we develop BANDIT, a Bayesian machine-learning approach that integrates multiple data types to predict drug binding targets. Integrating public data, BANDIT benchmarked a ~90% accuracy on 2000+ small molecules. Applied to 14,000+ compounds without known targets, BANDIT generated ~4,000 previously unknown molecule-target predictions. From this set we validate 14 novel microtubule inhibitors, including 3 with activity on resistant cancer cells. We applied BANDIT to ONC201—an anti-cancer compound in clinical development whose target had remained elusive. We identified and validated DRD2 as ONC201's target, and this information is now being used for precise clinical trial design. Finally, BANDIT identifies connections between different drug classes, elucidating previously unexplained clinical observations and suggesting new drug repositioning opportunities. Overall, BANDIT represents an efficient and accurate platform to accelerate drug discovery and direct clinical application.

[1] Institute for Computational Biomedicine, Dept. of Physiology and Biophysics, Weill Cornell Medical College, New York, NY 10065, USA. [2] Caryl and Israel Englander Institute for Precision Medicine, Weill Cornell Medical College, New York, NY 10065, USA. [3] Meyer Cancer Center, Weill Cornell Medical College, New York, NY 10065, USA. [4] Tri-Institutional Training Program in Computational Biology and Medicine, New York, NY 10065, USA. [5] Division of Hematology and Medical Oncology, Department of Medicine, Weill Cornell Medical College, New York, NY 10065, USA. [6] Oncoceutics, Inc., Philadelphia, PA 19104, USA. [7] Present address: OneThree Biotech Inc, Astoria, NY 11106, USA. [8] These authors contributed equally: Neel S. Madhukar, Prashant K. Khade. *email: josh.allen@oncoceutics.com; pag2015@med.cornell.edu; ole2001@med.cornell.edu

I t typically takes 15 years and 2.6 billion dollars to go from a small molecule in the lab to an approved drug[1–3], and for natural products and phenotypic screen derived small molecules, one of the greatest bottlenecks is identifying the targets of any candidate molecules[2,4]. Proper understanding of binding targets can position drugs for ideal indications and patients, allow for better analog design, and explain observed adverse events. There exist a number of experimental approaches for target identification ranging from affinity pull-downs to genome-wide knockdown screens[4,5], but these approaches are labor, resource, and time intensive, not to mention failure prone.

Computational approaches have the potential to substantially reduce the work and resources needed for drug target identification. Traditionally, ligand-based approaches take known binding targets for a given drug and attempt to find other drugs or proteins that are sufficiently similar[6]. However, to achieve high predictive power they require a large input of known binding partners for each tested drug, and therefore can only be used on drugs which have prior comprehensive target information[6,7]. Molecular docking, another commonly used approach, uses simulations of small molecules interacting with proteins to model if and how a drug may bind a given protein[8,9]. However, this approach requires significant computational power and complex 3D structures for each queried protein—data that is often unavailable.

Past work has used posttreatment gene expression changes and side effects to predict drugs new mechanisms for a given compound[10–14]. However, the majority of approaches rely on structural similarity between a queried compound and a database of drugs with known targets to predict new targets for the queried compound[15–17]. Yet, by relying on only a single data type these methods are more susceptible to data-specific noise and suffer from limited utility and accuracy. In addition, as new data types become more accessible and available, we expect single data-type methods to become less utilized by researchers.

Recently we have seen more methods emerging that attempt to integrate multiple different data types within a similarity based or data-driven framework[18–21]. However these approaches still suffer from a few limitations:

1. They use known targets of a given candidate compound as an input, which limits their applicability to orphan compounds with no known targets.
2. They often use gene-based similarity features, a method inherently biased against the discovery of diverse types of targets; favoring instead, the discovery of genes of the same class as the known drug-targets.
3. Most models only integrate one or two additional data types in addition to compound structure.
4. Many rely complex integration algorithms that are not easily able to accommodate new sources of information as they become available
5. Most have only evaluated their approach on a small number of drugs (<500) without thorough experimental validation.

To overcome these limitations, we introduce BANDIT, a drug-target prediction platform. BANDIT uses a Bayesian approach to integrate a number of diverse data types in an unbiased manner and provides a platform that allows for simple integration of new datatypes as they become available. Additionally, by integrating multiple different data types BANDIT is not reliant on any one experiment for its predictions and can achieve greater predictive power compared to single data type methods. Tested on ~2000 different compounds, BANDIT achieves a high accuracy at identifying shared target interactions, uncovers novel targets for the treatment of cancer, and can be used to quickly pinpoint potential therapeutics with novel mechanisms of action to accelerate drug development.

## Results

**An integrative approach leads to an increase in accuracy.** In the age of Big Data there has been an explosion of techniques that permit genomic, chemical, clinical, and pharmacological measurements to characterize a small molecule's mechanism. Many such measurements are either already published or are reasonably straightforward to perform. We hypothesized that integrating the multiple pieces of evidence provided by each data type into a cohesive prediction framework would dramatically improve target predictions. To test this hypothesis, we developed BANDIT: a Bayesian ANalysis to determine Drug Interaction Targets. BANDIT integrates over 20,000,000 data points from six distinct data types—drug efficacies[22], post-treatment transcriptional responses[10,11], drug structures[23,24], reported adverse effects[25], bioassay results[23,24], and known targets[26,27]—to predict drug–target interactions. This underlying database contains information on approximately 2000 different drugs with 1670 different known targets and over 100,000 unique orphan compounds (compounds with no known targets).

For each data type we calculate a similarity score for all drug pairs with known targets. Since each dataset uses a distinct reporting metric, the similarity calculation was specific to the data type being considered (Supplementary Fig. 1; "Methods"). Previous approaches have argued that high similarity in one feature indicates high similarity in others, implying that only one or two data types are sufficient for target prediction since others can be inferred[28]. However, using our expanded database, we found little overall correlation across different similarity scores (Fig. 1a; Supplementary Fig. 2). These results suggest that each data type is measuring a distinct aspect of a molecule's activity and further supported our hypothesis that integrating multiple data types could significantly improve target prediction accuracy.

We next separated drug pairs into those that shared at least one known target (~3% of all pairs) and pairs with no known shared targets. We applied a Kolmogorov–Smirnov test to each similarity score and used the associated $D$ statistic to calculate the degree a given data type could separate out drug pairs that shared targets (Fig. 1b). We found that all features were able to significantly separate the two classes ($P < 2e−16$), and structural similarity was found to be the most discriminative among all features evaluated ($D_{Structure} = 0.39$). In addition, we discovered that similarity across an unbiased set of bioassays and the relatively simple NCI-60 growth inhibition screen could strongly differentiate shared target drug pairs ($D_{Bioassay} = 0.327$ and $D_{GI50} = 331$), while, surprisingly[13,14,29], transcriptional responses ($D_{TResponse} = 0.1$) and reported adverse effects ($D_{SideEffect} = 0.14$) were much weaker differentiators. This information not only identifies the strengths of each data type, but will also allow researchers to efficiently prioritize experiments when faced with limited resources.

For every drug pair, BANDIT converts each individual similarity score into a distinct likelihood ratio. These individual likelihood ratios are then combined to obtain a total likelihood ratio (TLR) that is proportional to the odds of two drugs sharing a target given all available evidence (Fig. 1c; "Methods"). We chose to use a likelihood ratio approach because the ability to integrate available data (including newly generated data types) without a drastic change in protocol and the underlying interpretability in identifying how individual features contribute to a given prediction. We calculated TLRs for all possible drug pairs with known targets and the output was evaluated using 5-fold cross validation. We observed an Area Under the Receiver Operating Curve (AUROC) of 0.89 demonstrating that

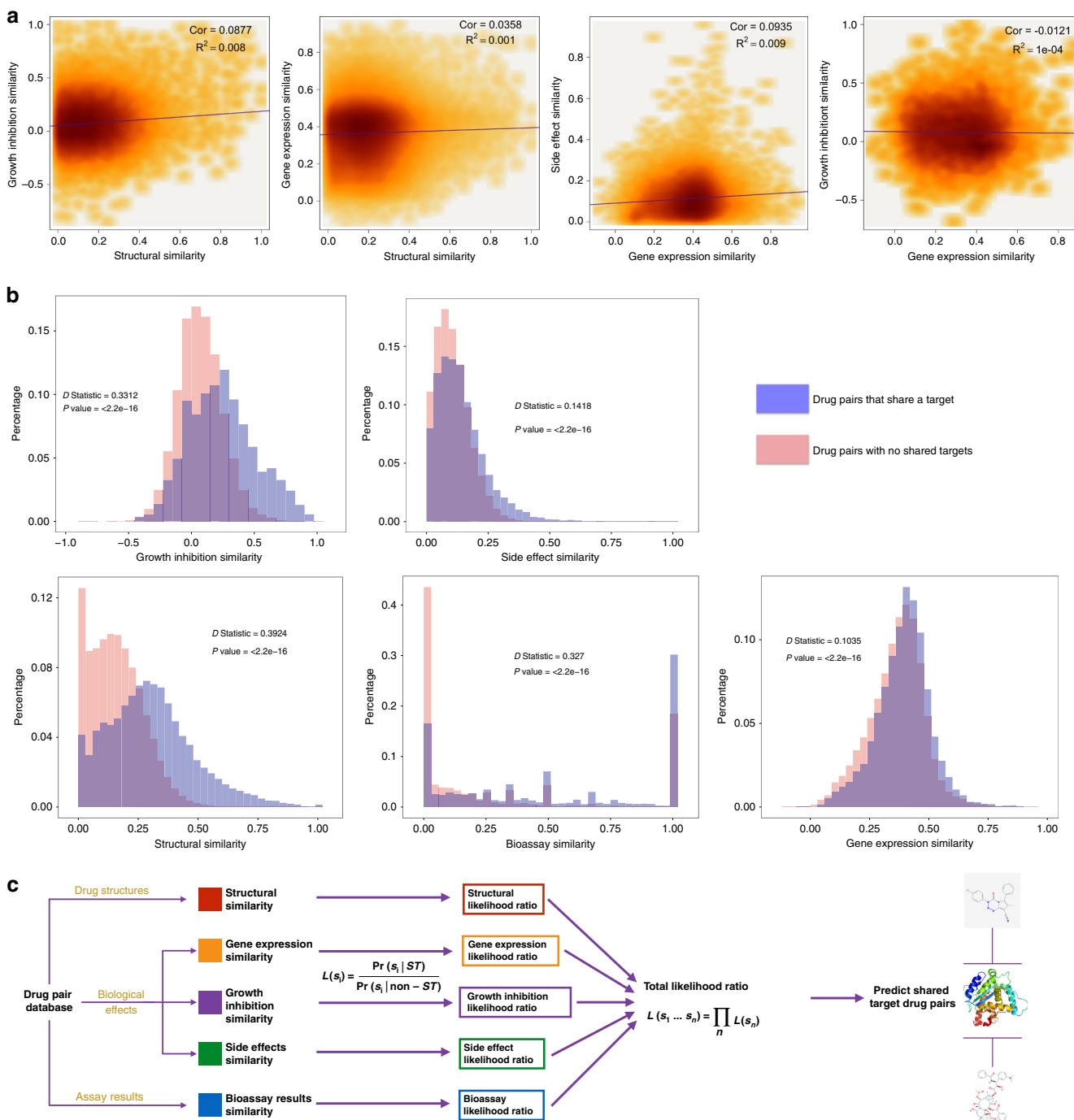

**Fig. 1** BANDIT exploits the individual predictive powers of each data type. **a** Density plots showing how various different similarity scores correlate with one another, with darker area corresponding to a higher density of values. $R^2$ and $P$ value were calculated using a pearson correlation. **b** Distributions of similarity scores across two sets—drug pairs known to share a target and those with no known shared targets. $P$ values and $D$ statistics were calculated using the Kolmogorov–Smirnov test. **c** Schematic of BANDIT's method of integrating multiple data types to predict shared target drug pairs

BANDIT's integrative approach can accurately identify drugs that share targets. To further test this, we compared the ROC curve and Precision-Recall curve of BANDIT to one where we randomly shuffled the likelihood values (representing what you would expect if BANDIT's TLR had no predictive power), and saw an improvement compared to random in both tests (Supplementary Figs. 3, 4). We then recomputed the AUROC while varying the number of included data types and observed an overall increase in predictive power as we added new data types, regardless of the addition order (Fig. 2a, Supplementary Table 1).

As expected, we observed that the variables that had the greatest increase in AUROC were the same variables that showed the greatest separation in the KS tests. This result verified the power of BANDIT's Big Data approach and demonstrated how separate information sources can be combined to yield predictions more powerful than those obtained from any individual source (Supplementary Fig. 5). This was confirmed using the KS test where we saw that the TLR output could better separate shared target drug pairs than any individual similarity score with a drastic increase in performance when focusing on drug pairs with

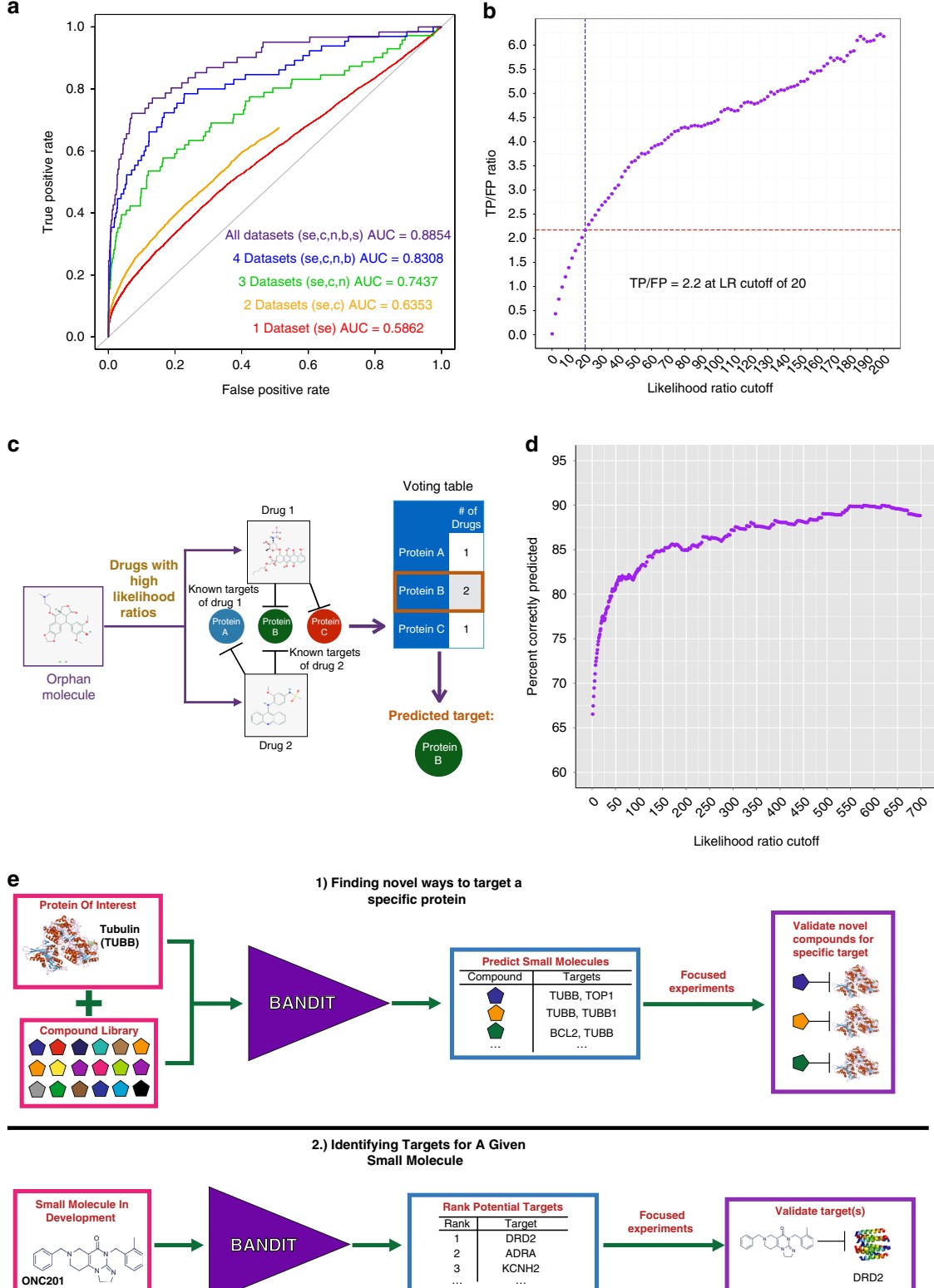

**Fig. 2** BANDIT can accurately predict shared targets and specific target interactions. **a** Area under the receiver-operating curve for different sets of data types. SE = Side effects; C = CMap; N = NCI60; B = Bioassays; S = Structure. **b** Ratio of true positives to false positives at different likelihood ratio cutoffs. **c** Schematic of the BANDIT voting schematic for predicting specific target interactions. **d** Accuracy level of BANDIT's voting algorithm at various likelihood ratio cutoffs. **e** Schematic of two proposed operating scenarios for BANDIT

all 5 data types ($D_{TLR} = 0.69$, Supplementary Fig. 6). Furthermore, we observed that BANDIT's ratio of true to false positives continually increased as we raised the cutoff value, indicating that BANDIT can effectively pick out high quality shared-target predictions (Fig. 2b, Supplementary Fig. 7). In addition we found that BANDIT could pinpoint known shared target drugs better than many existing shared target prediction methods ("Methods").

**BANDIT accurately predicts specific target interactions**. We next investigated how we could use BANDIT to replicate results from published experimental screens. Peterson et al.[30] tested 178 known protein kinase inhibitors against a panel of 300 different kinases and measured the level of inhibition (in terms of percent remaining kinase activity) for each inhibitor–kinase pair. We examined all orphan molecules—molecules with no known targets—in both the Peterson kinase database and BANDIT's, and, used BANDIT to predict potential kinases targets for each orphan molecule ("Methods"). We observed that the kinase targets BANDIT predicted for each orphan molecule had higher levels of reported inhibition in the Peterson dataset than non-predictions ($p < 1e-5$; Supplementary Fig. 8). This result highlights how we could use BANDIT to guide experimental screens while minimizing operational costs.

Moving forward from shared-target predictions, we examined whether for a given drug BANDIT could be used to predict a specific binding target. We hypothesized that if a protein appeared as a known target in a large number of shared target predictions, then it is likely a target for the tested orphan molecule. To test this hypothesis, we developed a voting algorithm to predict specific targets for each orphan small molecule by identifying recurring targets (Fig. 2c, "Methods"). We applied our voting method to all drugs in our database with known targets and observed the accuracy level—measured by whether BANDIT correctly identified a known drug target—steadily increased as we raised the TLR cutoff for a shared target prediction before reaching an overall accuracy of ~90% (Fig. 2d). This demonstrated that BANDIT could be used to accurately identify specific targets for a diverse set of small molecules.

We then used BANDIT to predict novel targets for 14,000+ small molecules with no known targets or mechanisms of action in our database. Each of these molecules had data available in at least three of the five data types considered by BANDIT. We confidently predicted targets for 4167 unique small molecules (30% of our original set), with predictions spanning over 560 distinct protein targets. By setting a higher TLR cutoff for predictions and requiring a higher number of votes for any predicted targets, we further narrowed this list to 720 high confidence target predictions. Based on this, we envisioned two main operating scenarios for BANDIT: (1) Using BANDIT in combination with a library of orphan small molecules to identify new small molecules targeting a specific protein and (2) to integrate BANDIT directly into the drug development pipeline to predict targets and guide experiments for drugs currently in development (Fig. 2e).

**Discovery of novel microtubule-targeting compounds**. Beginning with the first operating scenario, we used BANDIT to identify novel ways to target microtubules. Antimicrotubule drugs make up one of the largest and most widely used classes of cancer chemotherapeutics, with tubulin being one of the most validated anticancer targets to date[31–34]. Interestingly, and unlike most classes of chemotherapy drugs or targeted-therapies in oncology, microtubule inhibitors are further sub-categorized as microtubule-stabilizing (e.g. taxanes) and microtubule-

depolymerizing drugs (e.g. vinca alkaloids). Each class shifts the cellular equilibrium that normally exists between soluble tubulin dimers and microtubule polymers, towards microtubules (taxanes) or soluble tubulin (vinca alkaloids). Despite the clinical success of the entire class of microtubule inhibitors, the development of drug resistance—which is the number one cause of cancer mortality in metastatic patients—limits their clinical applicability[35]. Hence, the discovery of novel microtubule-targeting small molecules could significantly improve cancer therapy by identifying compounds with activity on refractory tumors. To this aim, we further focused our list of high confidence orphan-target predictions to small molecules predicted to target microtubules. To see how our novel predictions related to known microtubule-targeting therapeutics, we created a network of all known and predicted antimicrotubule small molecules with edges representing a predicted shared target interaction (Supplementary Fig. 9). Interestingly we found that the known microtubule-targeting agents tended to cluster together based on their distinct mechanism of action. For instance, we observed Paclitaxel clustering with Cabazitaxel and Docetaxel—all known microtubule-stabilizing drugs—while Colchicine clustered with other known microtubule-destabilizing drugs such as Podophyllotoxin. This is especially exciting since it demonstrates the potential for BANDIT to be used not only to identify a specific target for an orphan molecule but to differentiate between different modes of action on the same target.

From our list of top antimicrotubule drug predictions (TLR > 100) we obtained a set of 24 compounds with varying structures for experimental testing ("Methods", Supplementary Table 2). We chose human breast cancer MDA-MB-231 cells for the validation experiments as microtubule-inhibitors (both stabilizing and destabilizing) are commonly used in the treatment of breast cancer patients. Cells were treated for 6 h with 1 and 10 μM of each small molecule, and the integrity of the microtubule cytoskeleton (assessed by confocal microscopy following tubulin immunofluorescence) was used as the bioassay endpoint. Additionally, to identify any compounds that may have poor cellular permeability, we performed a crude-tubulin assay in which cell were lysed in the presence of 10 μM of each compound for 30 min before separating polymerized from soluble tubulin by centrifugation. Our results showed that 14 of the 24 orphan small molecules exhibited significant effects on microtubules (Fig. 3a–f, Supplementary Figs. 10–15, Supplementary Table 2), with 13 compounds showing success in both the cell-based and crude-tubulin assays, and one compound (#3) showing activity only in the crude-tubulin assay. Overall, this represents a much higher success rate (58%) than one would expect by chance ($p < 2e-16$, "Methods"). In addition, we found that only nine out of these confirmed 14 hits would have been predicted to bind microtubules by drug target prediction methods that relied only on compound structure[16,17]. To more accurately quantify the extent of drug–target engagement, we employed a second biochemical assay quantifying the effect that each small molecule exerted on the equilibrium between microtubule polymers and soluble tubulin, following 6 h of treatment (Supplementary Fig. 16). Our results confirmed and corroborated the microscopy results, further revealing that while several small molecules had maximal microtubule-inhibitory activity at the lowest dose (1 ⌈M) (Fig. 3c–f), others exhibited a dose-dependent effect on microtubule depolymerization (e.g. compounds #12, #13), further establishing microtubules as their bona-fide target (Fig. 3g–i). Taken together, these experiments confirmed the predicted targets and mechanism of action for the majority of the newly identified microtubule inhibitors. While further testing will be needed before these small molecules can be used

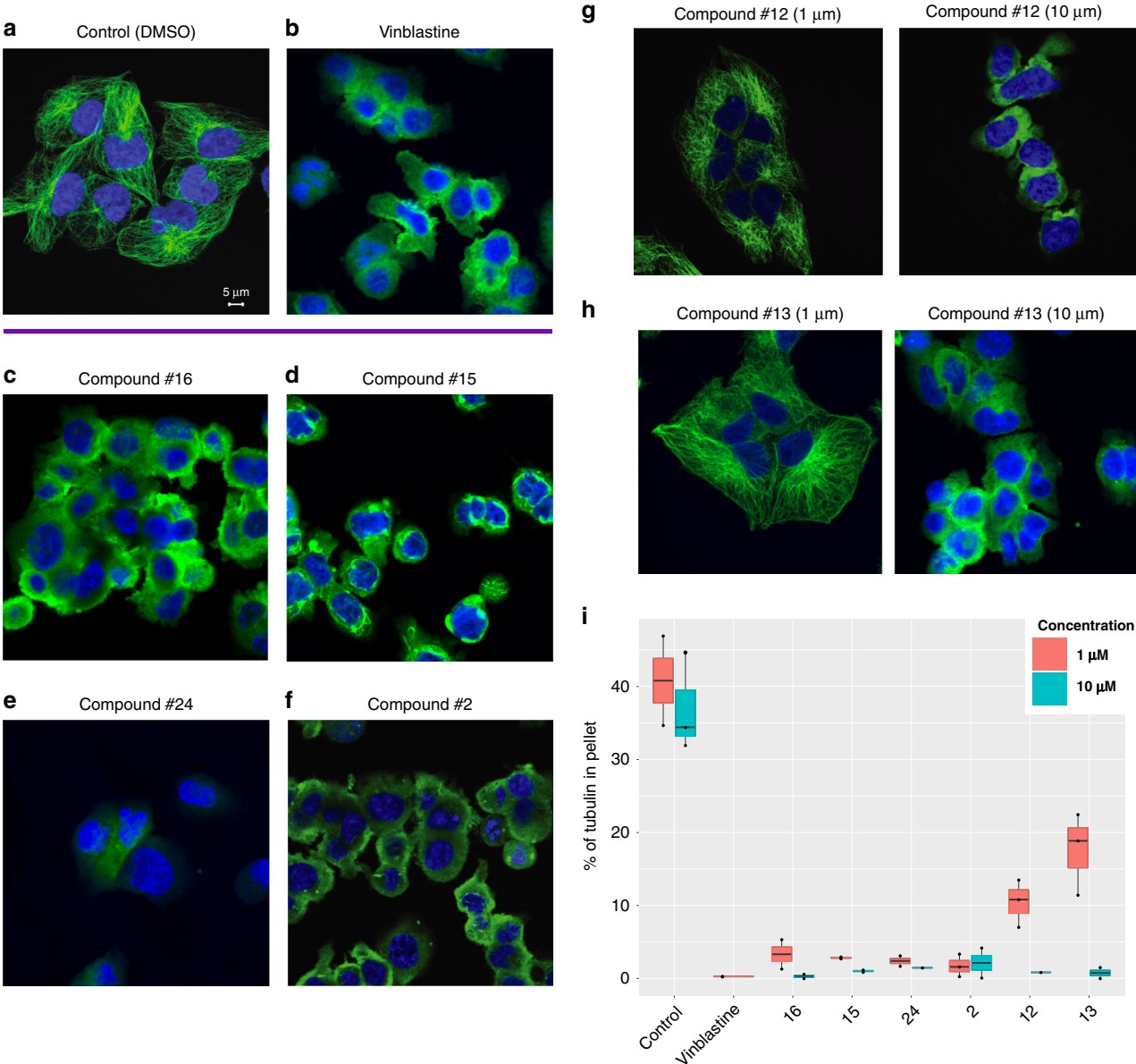

**Fig. 3** Microtubules are a correct target of the newly identified small molecules. Effect of various compounds (1 μM) on the microtubule integrity of MDA-MB-231 cells after 6 h of treatment. **a** Control with DMSO (Scale bar: 5 μm), **b** Vinblastine as a positive control, **c** Compound #16, **d** Compound #15, **e** Compound #24 **f** Compound #2. **g** Dose dependent effect of Compound #12 and **h** Compound #13. **i** Box plot showing the % tubulin in the pellet compared to the supernatant for depolymerizing drugs at 1 and 10 μM. The median is denoted by the center line and the min/max are represented by the whiskers. Each individual replicate is represented by a point in the box plot

clinically, these results do demonstrate how BANDIT can be used on compound libraries to identify small molecules acting on specific targets.

To inform future clinical development for these newly identified microtubule inhibitors, we next tested their activity against resistance models. In the case of microtubule inhibitors, overcoming drug resistance is especially challenging as the mechanisms are often multifactorial. As previously demonstrated, BANDIT can accurately identify a set of structurally diverse small molecules that all bind a common target (in this case microtubules), therefore we next investigated whether any of our newly identified microtubule-depolymerizing small molecules could successfully act on tumors resistant to other known antimicrotubule drugs. Using the 1A9 human ovarian carcinoma cell line—which has previously been used successfully in selecting microtubule-inhibitor resistant clones and for high throughput

small molecule screening,[36–40]—we created clones resistant to Eribulin mesylate, a microtubule depolymerizing drug that is FDA approved for the treatment of docetaxel-refractory breast cancer patients[41,42] (Fig. 4a). Interestingly, recent clinical data demonstrated that fewer than 50% of breast cancer patients showed any detectable response after treatment with Eribulin, further highlighting the importance of finding new molecules that share the same validated target but are active against the large population of refractory patients[43]. Using 72-h growth inhibition assays we observed that the Eribulin-resistant 1A9 cells (1A9-ERB) were more than 7000 fold more resistant to Eribulin than the parental cells and exhibited cross-resistance to all classes of clinically used microtubule-depolymerizing drugs (Supplementary Table 3). To test whether the drug-resistance phenotype was due to impaired drug–target engagement, we treated parental and resistant cells for 6 h with 1 μM of Eribulin or each of the FDA-

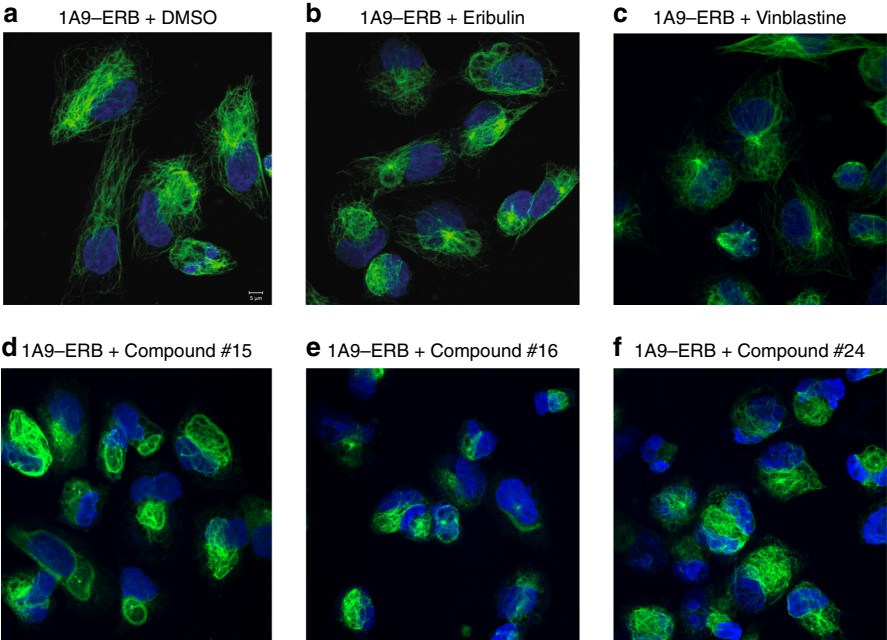

**Fig. 4** BANDIT predicted small molecules can act on resistant cells. Effect of various compounds on the microtubule integrity of 1A9-ERB cells after 6 h of treatment: **a** Control with DMSO (Scale bar: 5 μm), 100 nM of **b** Eribulin and **c** Vinblastine, and 1 μM of **d** Compound #15, **e** Compound #16 and **f** Compound #24

approved depolymerizing drugs. Consistent with their drug resistance phenotypes, our results showed lack of drug-induced microtubule depolymerization in 1A9-ERB cells in contrast to the complete depolymerization observed in the microtubule network of drug-sensitive 1A9 parental cells (Fig. 4b, c, Supplementary Figs. 17, 18). These on-target drug efficacy results are in agreement with the lack of anticancer activity revealed by the growth inhibition data, further highlighting the importance of discovering novel small molecules that could act on these refractory tumors. We tested the top four performing small molecules (#15, 16, 24, and 2) on the 1A9-ERB cells and found that three out of four compounds tested, were active against the 1A9-ERB cells and effectively depolymerized microtubules, as evidenced by the diffuse soluble tubulin staining following drug treatment (Fig. 4d–f, Supplementary Figs. 17, 18), in contrast to the fine and intricate microtubule network observed in untreated cells (Fig. 4). Compound #15, which was the most active of the four compounds, was tested using growth inhibition assays and was found to almost completely circumvent drug-resistance from 7050-fold observed with Eribulin down to 4-fold (Supplementary Table 3). While further in vitro and in vivo studies are required for the clinical development of these compounds, these results clearly demonstrate BANDIT's utility in identifying lead small molecules with potential activity against drug resistance tumor models without the labor- and cost-intensive physical screening of thousands of small molecules. Compounds such as these could represent the next generation of clinically developed drugs reducing the need for extensive medicinal chemistry and structure–activity studies.

**BANDIT uncovers ONC201's selective antagonism of DRD2.**
Given BANDIT's demonstrated capability to accurately identify specific targets for orphan small molecules, we next investigated how we could integrate BANDIT directly into the drug development pipeline and test its ability to predict targets for small molecules with promising clinical activity but without a specific target. Therefore we applied BANDIT to ONC201—an orphan small molecule discovered through a phenotypic screen for p53-

independent inducers of TRAIL-mediated apoptosis—currently in multiple phase II clinical trials for select advanced cancers. Despite its promising preclinical and early clinical anticancer activity and its reported effects on a few signaling pathways, including Akt/ERK pathway[44–46], a bona-fide target for this compound remained elusive.

To identify direct binding targets for ONC201, we used BANDIT to compute likelihood ratios between ONC201 and all drugs with known targets in BANDIT's database. BANDIT's top shared target prediction were between ONC201 and Oxiperomide and Thioridazine, both dopaminergic antagonists previously used the treatment of dyskinesias and schizophrenia respectively[47–50]. Interestingly, our voting analysis indicated that the most likely targets of ONC201 were dopamine receptors—specifically DRD2 —and adrenergic receptor alpha (Fig. 5a), both of which are members of the G-protein coupled receptor (GPCR) superfamily. Further highlighting the novelty of these predictions, we found that DRD2 was not predicted as a target of ONC201 by other commonly used target prediction algorithms (such as SEA and SuperPred)[15,16].

To test these predicted targets we performed in vitro profiling of GPCR activity using a hetereologous reporter assay for arrestin recruitment, which is a hallmark of GPCR activation[51]. Our results indicated that ONC201 selectively antagonized the D2-like (DRD2/3/4), but not D1-like (DRD1/5), subfamily of dopamine receptors (Fig. 5b; Supplementary Fig. 19A), with no observed antagonism of other GPCRs under the evaluated conditions. Among the DRD2 family, ONC201 antagonized both short and long isoforms of DRD2 and DRD3, with weaker potency for DRD4. Further characterization of ONC201-mediated antagonism of arrestin recruitment to DRD2L was assessed by a Gaddam/Schild EC50 shift analysis, which determined a dissociation constant of 2.9 μM for ONC201 that is equivalent to its effective dose in many human cancer cells (Fig. 5c). Confirmatory results were obtained for cAMP modulation in response to ONC201, which is another measure of DRD2L activation (Fig. 5d). The ability of dopamine to completely reverse the dose-dependent antagonism of up to 100 μM ONC201 suggests direct, competitive

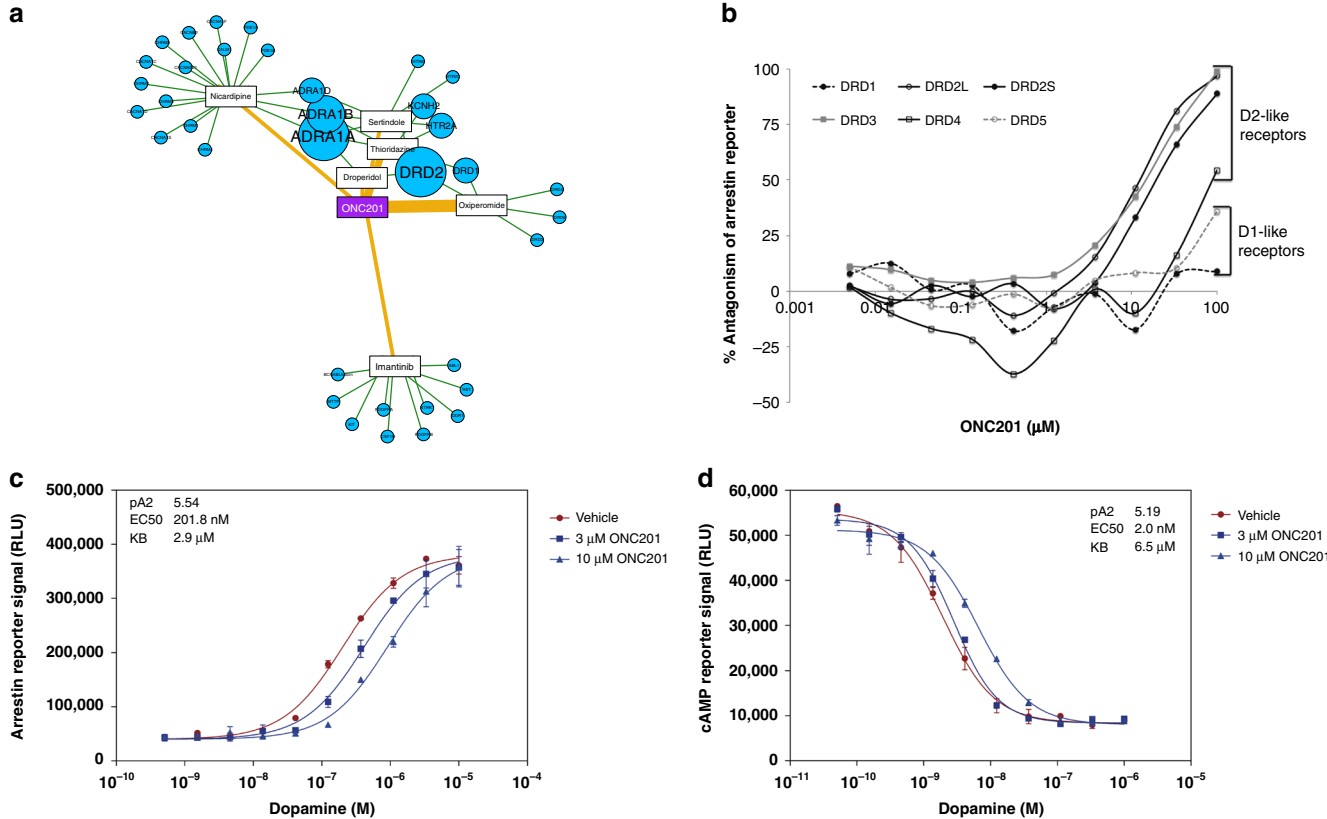

**Fig. 5** ONC201 is a selective DRD2 antagonist. **a** BANDIT target predictions for ONC201. Connections between ONC201 and known drugs are weighted based on the likelihood ratio and predicted targets are sized based on the prediction strength. **b** Antagonism of ligand-stimulated dopamine receptors by ONC201. **c** Schild analysis of DRD2L antagonism by ONC201 using arrestin recruitment or **d** cAMP modulation reporters. Error bars represent 1 standard deviation

antagonism of DRD2L (Supplementary Fig. 19B, C). In agreement with the specificity of ONC201 for the target predicted by BANDIT, no significant interactions were identified between ONC201 and nuclear hormone receptors or the kinome (Supplementary Fig. 19D, E). Interestingly, a biologically inactive constitutional isomer of ONC201[52] did not inhibit DRD2L, suggesting that antagonism of this receptor could be linked to its biological activity (Supplementary Fig. 19F). Additionally, since this discovery (first published in a preprint of this work)[53] follow-up publications have further elucidated the role of DRD2 in the anticancer activity of ONC201 through knockout, efficacy based, and clinical studies[54,55]. In summary, these results showed that ONC201 selectively antagonizes the D2-like subfamily of dopamine receptors, which is an unconventional target for oncology drugs and further demonstrate BANDIT's ability to act as a tool to advance drug development.

This unexpected discovery on the DRD2 being a direct-binding target for ONC201, has also led to the design and launch of a clinical trial of ONC201 in pheochromocytomas, owing to high levels of DRD2 expression in this rare tumor type (trial identifier: NCT03034200). Taken together, these results demonstrate the extreme potential of BANDIT to expedite drug development by using global, novel drug-target engagement predictions in combination with gene expression studies to enable the identification of select patient and indications groups more likely to benefit from a particular drug treatment.

**BANDIT can identify specific drug mechanisms.** Following validation that BANDIT could accurately determine the specific targets for small molecules, we then examined how it could also

be used to understand the target binding mechanism, otherwise known as its mechanism of action (MoA). First we used BANDIT to test all known microtubule-targeting drugs, and created a hierarchical cluster based on their TLR outputs ("Methods"). We observed a clean separation between known microtubule depolymerizing and polymerizing agents (Fig. 6a). A similar MoA-based clustering was observed when we tested all known protein kinase inhibitors, which showed a clear separation between receptor tyrosine kinase inhibitors, serine/threonine kinase inhibitors, and nucleoside analogs (Fig. 6b). Overall these results demonstrate that BANDIT can be used to differentiate small molecules based on their specific MoA without additional model training. Combined with the earlier voting algorithm, this demonstrates an efficient pipeline for small molecule target and mechanism identification: first using BANDIT to predict targets for an orphan small molecule, followed by clustering with other drugs known to act on the same target to discern MoA.

**Identifying connections within the drug universe.** We next used BANDIT to get an overview of how different classes of drugs, spanning the entire clinical landscape, may be related to one another. Based on the TLR between each drug pair, we constructed a network representative of the drug universe, or all known drugs with at least one predicted shared target interaction (Fig. 6c). Each drug was classified according to its 1st order Anatomical Therapeutic Chemical (ATC) classification—characteristic of the type and intended use of each drug. As expected, drugs of a similar ATC code cluster together, however we also observed many unexpected clusters indicative of drug mechanisms or effect. Interestingly, among all classes of

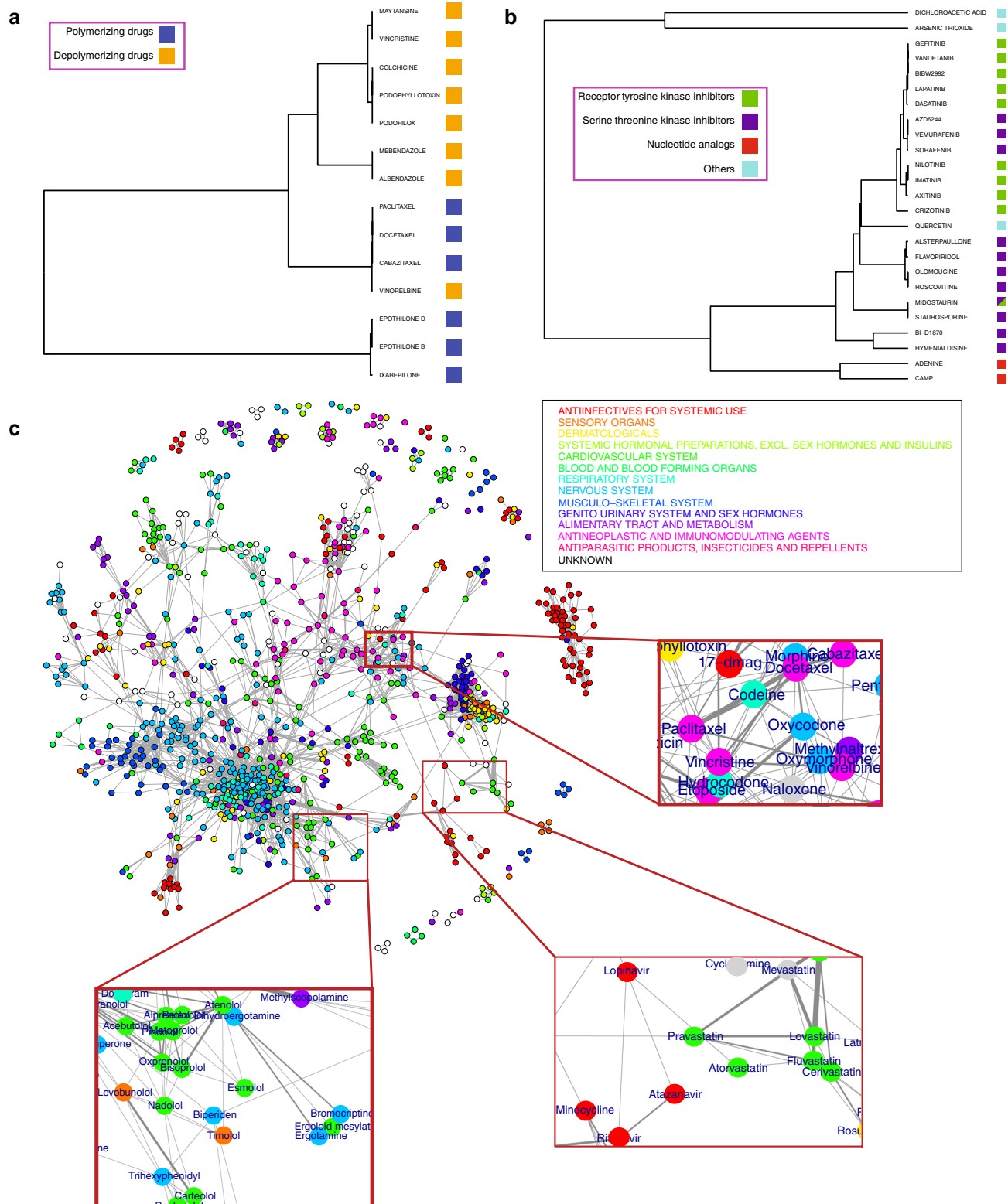

**Fig. 6** BANDIT can predict specific mechanisms of action and connections between drug classes. **a** Hierarchical clustering of drugs known to target microtubules and **b** drugs known to target protein kinases. **c** Network of drugs based on shared target interactions. Drugs are colored based on their most prevalent ATC code. Three specific clusters corresponding to beta-blockers and Parkinson's medications, anti-retrovirals and statins, and opioids and antimicrotubule drugs are highlighted

chemotherapeutics, microtubule inhibitors clustered together with camptothecin analogs, for which a dual role as topoisomerase I and tubulin polymerization inhibitors has been previously reported[56], but which is not widely acknowledged in clinical oncology. Conversely, we unexpectedly found opioids closely interconnected with microtubule targeting agents; this unanticipated observation is in line with previous reports showing how exposure to microtubule targeting drugs can increase the levels of the opioid receptor in rat cerebellums and that treatment of cardiac myocytes with opioids induces microtubule alterations[57,58]. This unexploited finding could reveal novel biology linking the opioid receptor signaling pathway with the microtubule cytoskeleton, as well as potentially represent an example of drug repurposing, suggesting novel clinical indications for drugs already FDA-approved. As further proof of the clinical value of the broad universe clustering information revealed by BANDIT, we detected close clustering of known beta-blockers with many Parkinson's medications, which was especially interesting given that one of the most controversial clinical applications of beta-blockers was to reduce tremors in Parkinson's patients[59]. Drug clustering was also strongly indicative of potential side effects, as suggested by the link between antiretroviral medications, which often cause metabolic side effects like hypercholesterolemia, and statins, FDA-approved cholesterol lowering drugs[60]. Overall we believe this broad universe clustering approach could greatly advance future drug development by indicating novel potentially synergistic drug combinations, potentially cumulative side effects, and by assisting in drug repositioning.

## Discussion

We have developed BANDIT, an integrative Big-Data approach that combines a set of individually weak features into a single reliable and robust predictor of shared-target drug relationships and individual drug binding targets. Our predictions replicated shared-target relationships, individual drug–target relationships, and known mechanisms of action within our test set and replicated results of large-scale experimental screens. Moreover, we experimentally confirmed several of our novel predictions using different bioassays and model systems and demonstrated BANDIT's capability to efficiently discover novel small molecules, which could be used in refractory tumors. Finally, BANDIT can be used on a broader scale to discern mechanisms of approved drugs, characterize the global drug universe landscape, explain existing, yet puzzling, clinical phenotypes, and repurpose drugs for new indications.

In addition, as BANDIT can be continually augmented with new and diverse datasets, we expect its predictive power to increase as new data becomes available. This is due to one of the key strengths of the Bayesian framework, as it can easily accommodate new features and quickly evaluate their individual predictive power by computing direct probabilities. However, as more information becomes available there are many aspects of the current implementation that can be improved. For instance, we can better understand the dependencies between distinct data types and model those within our Bayesian network, and as more information on binding kinetics becomes available, BANDIT could be adapted to better predict on versus off-target effects. As drug development often stops in early clinical studies due to unanticipated toxic side effects, BANDIT could help overcome these roadblocks by identifying side effects due to unknown off-target bindings.

In summary, we show herein the potential of BANDIT in expediting drug development, as it spans the entire space ranging from new target identification and validation to clinical candidate

development and drug repurposing. By allowing researchers to quickly obtain target predictions it could streamline all subsequent development efforts and save scientists both time and resources. Furthermore, BANDIT could be used to rapidly screen a large database of compounds and efficiently identify any promising therapeutics that could be further evaluated. Overall our results demonstrate that BANDIT is a novel and effective screening and target-prediction platform for drug development and is poised to positively impact current efforts.

## Methods

**Datasets**. Growth inhibition data: We used publicly available growth inhibition data from the National Cancer Institute's Development Therapeutics Program (NCI-DTP). Each of the NCI60 cell lines were treated with a small molecule and the concentration that caused a 50% decrease in cell growth was measured. When there were multiple high quality experiments done for the same compound, we averaged the values to obtain a single GI50 value for each small molecule—cell line pair. Contains data on 20,000+ unique compounds. Version 1.6.2 was downloaded from cellminer.com.

Gene expression data: All post-treatment gene expression data was downloaded from the Broad Connectivity Map (CMap) project. Fold change data across all cell lines were averaged to obtain a single gene expression signature for each compound. Contains data on 1309 different compounds. Build 02 was downloaded from the Broad CMap Portal.

Adverse effects: Side effects (mined from drug package inserts and public information) were downloaded from the SIDER database. Each side effect was classified using the MedDRA (version 16.1) dictionary.

Bioassays/Chemical structures: All bioassay results and chemical structures were downloaded from PubChem and organized based on each small molecule's PubChem Compound Identification (CID).

Known Drug Targets: All known drug targets were extracted from the DrugBank database (Version 4.1).

**Calculating similarity scores**. Growth Inhibition Data: For each pair of drugs we calculated a pearson correlation value across the 60 data points (Supplementary Fig. 1).

Gene expression and Chemogenomic Fitness Scores: A pearson correlation was used to measure the degree of similarity for the profiles of two drugs

Bioassays: All bioassays were classified as either positive or negative based on the data available in Pubchem. A jaccard index was calculated based on the number of shared "positive" assays between two drugs. We required that each drug pair have been tested in at least one similar assay for a similarity score to be calculated.

Chemical Structures: For each drug we extracted the isomeric SMILES and used the atom-pair method[61] to calculate the DICE coefficient based structural similarity between two compounds (Supplementary Fig. 1). Other structural similarity methods (such as ECFP with Tanimoto coefficients) were evaluated as well, however all had a lower overall predictive power—measured by the D-statistic from a KS test—therefore the DICE similarity was ultimately chosen.

Adverse Effects: Using the SIDER2 database[25] we extracted the "preferred term" side effects for each drug. A jaccard index was then calculated for the shared side effects for each drug pair.

**Calculating correlations between similarity types**. To combine data from different databases, we mapped information from each drug back to a PubChem compound identifier (CID) that was used for all subsequent integration. For each pair of similarity scores we separated out drug pairs where both similarity types were measured and plotted the different similarity scores against one another (Fig. 1a, Supplementary Fig. 2). We computed the Pearson correlation coefficient (PCC) and the coefficient of determination ($R^2$) between each pair of similarity scores. Across all pairs, we observed a low correlation—measured by both the PCC and $R^2$. This finding demonstrated that high similarity of one type does not necessarily implied high similarity in another.

**Calculating the total likelihood ratio**. For each data type BANDIT calculates a likelihood ratio $L(s_n)$ is defined as the fraction of drug pairs with a shared target (ST pairs) having a given similarity score $s_n$, divided by the fraction of the non-ST pairs with the same similarity score:

$$L(s_i) = \frac{\Pr(s_i|\text{ST})}{\Pr(s_i|\text{non-ST})} \quad (1)$$

For each data type we binned similarity scores into 20 evenly spaced intervals and calculated the likelihood value for all similarity scores in each bin. We then fit an exponential function (using the R "predict" and "exp" methods) to each data type, and this was used to calculate likelihood values for new cases.

Our previous analysis highlighted the minimal correlation between the similarity types and how data types could be modeled using a Naïve Bayes framework. This implies that the joint probability of two drugs sharing a target

given a set of similarity scores can be modeled as the product involving individual similarity scores. Overall we decided to use this Bayesian framework for multiple reasons, such as the readily interpretable nature of a likelihood ratio compared to other more complicated machine learning scores and the ability to easily add in new data types as they become available.

Therefore the total likelihood ratio $L(s)$ can be expressed as the product of the individual likelihood ratios:

$$\text{TLR} = L(s) = \prod_n L(s_{1-n}) = L(s_1)L(s_2)...L(s_n)$$

$$n = maximum\ \#\ of\ included\ datasets \tag{2}$$

The total likelihood ratio (TLR) is then proportional to the odds of two drugs sharing a given target $n$ given sources of information. If a data type is not available for a given compound then the median value of all similarity scores for that data type was used to calculate the likelihood value. This imputation was done after the similarity to likelihood conversion was established (Eq. 1) so as not to skew likelihood values.

**Testing against drugs with known targets**. Drug targets were extracted from DrugBank and drug pairs were classified as a "shared-target" pair if they had at least one target in common. We used fivefold cross validation to split our set of drug pairs into a test and training set containing 20% and 80% of the drug pairs respectively. We sub-sampled the two classes (ST and non-ST drug pairs) and required the ratio of true positives (ST pairs) to true negatives (non-ST pairs) to remain the same as the total set. For each fold we computed TLRs for each drug pair in the test set based on the background probabilities within the training set. Each of the five test folds combined at the end to produce an ROC Curve and calculate the AUROC value. We calculated the AUROC value for each individual likelihood ratio from a single data type (Supplementary Fig. 5).

We performed this analysis with the TLR output while varying the number of data types being considered and found a significant increase in the predictive power, measured by the AUROC, as we increased the number of included datasets (Fig. 2a). We computed two sets of ROC curves—one where we required drugs have available data in each included data type (our preferred method) and another where we imputed the data type median for each missing data type. We varied the order in which datasets were added and observed a positive relationship between AUROC value and the number of included data types regardless of the addition order. We tested this by selecting each possible combination of the five data types and computing the AUROC using five-fold cross validation and observed an increase in the average AUROC as the total number of included data types increased (Supplementary Table 1). Furthermore, we used a KS test to measure how our TLR value could separate out ST and non-ST pairs and saw that in each case our TLR value outperformed any individual variable (Supplementary Fig. 6). We repeated this analysis increasing the minimum number of data types we required a pair of compounds to have and saw the separation steadily improve ($D = 0.44$–$0.69$).

**Comparison to existing drug target prediction methods**. Campillos et al.[13]: In their paper "Drug Target Identification Using Side-Effect Similarity," Campillos et al. report that they used their method to test 246,051 drug pairs (in which 6681 were known to share targets) to predict shared target relationships[13]. In comparison to BANDIT, this method relied on only side effect similarities to identify shared target drugs. They report that out of their top 409 predictions (representing the top 166% of all predictions), 192 were known to share targets (representing an accuracy of 47%). When we performed a similar test on our top 166% of predictions (2160 drugs pairs out of 1,301,691 total), we found that 1649 were known to share targets. This represents a 76% accuracy for top predictions—a significant improvement from the 47% reported by Campillos et al. To further evaluate our comparative accuracy, we also looked deeper to the top 1.2% of predictions. In their paper, Campillos et al. report that out of their top 2903 predictions (representing the top 1.2%), 956 were known to share targets (representing a 33% accuracy). When we evaluated our top 1.2% of predictions (15,620 drug pairs) we found that 6,884 were known to share at least one target (44% accuracy), once again highlighting BANDIT's increased accuracy to established methods (Supplementary Table 4).

Wang et al.[14]: In their paper "Prediction of Drug–Target Interactions for Drug Repositioning Only Based on Genomic Expression Similarity" Wang et al. report an AUC of 0.66 when predicting shared target relationship of compounds in the CMap database using the BAES method, which integrates batch corrected gene expression information. Using BANDIT we observed a higher AUC when evaluated on all compounds in our database that were tested by CMap (Supplementary Table 4).

Caniza et al.[62]: In their paper "Mining the biomedical literature to predict shared drug targets in DrugBank" Caniza et al. compared multiple similarity driven approaches based on prior biomedical literature to identify how well they could identify drugs which share at least one target (AUC = 0.51–0.69). Each of these methods uses semantic similarity approaches to find shared target drugs, but only integrate drug MESH terms or chemical structures. We found that BANDIT outperformed each of these methods when tested on all drugs (Supplementary Table 4).

**Replicating kinase experimental screen**. We first separated out the kinases in the Peterson et al. database[30] that were classified as BANDIT orphan small molecules —molecules that were in at least two of the considered BANDIT databases and had no known targets (11 unique compounds). For each orphan kinase inhibitor we used BANDIT to predict shared target drugs. Each known kinase target of the shared target drugs was classified as a potential kinase target of the orphan inhibitor. We then observed that the "percent remaining kinase activity" was significantly lower between the orphan kinase inhibitors and the BANDIT predicted kinases than between the orphan inhibitors and any non-predicted kinases (Wilcoxon Rank Sum Test $P = 3.62e{-}06$) (Supplementary Fig. 8).

**Specific target voting**. For each orphan small molecule, we identified all shared target drug predictions, or any drugs with known targets that exceeded a given BANDIT likelihood ratio. For each shared target drug prediction, we compiled all known targets of that given drug and ranked specific protein targets based on how often it appeared as known target in shared drug target predictions. Votes for particular protein targets were weighted based on the likelihood ratio of the shared target prediction they originated from. The top voted target for each orphan small molecule that we tested was then predicted to be a novel specific target (Fig. 2e).

To test the accuracy, we used leave-one-out cross validation on our test set of drugs with known targets. For each drug we used BANDIT to compare it to all other drugs with known targets and identify the top ranked target for the tested drug. This was repeated for every drug in our test set and we calculated how often the top ranked target was a known target of the drug being tested. We recomputed these accuracies while varying the likelihood ratio cutoff for a drug pair to be considered a shared-target prediction. As expected we observed a steady increase in accuracy as we increased the cutoff value, with the accuracy plateauing at an accuracy level of ~90%—revealing that BANDIT's voting protocol could accurately identify specific targets (Fig. 2f). We also redid this test using fivefold cross validation and observed little difference in overall accuracy (plateauing at ~87% accuracy).

**Identification of novel antimicrotubule small molecules**. For each orphan small molecule in BANDIT (defined as a molecule tested in any of the individual databases but without any known targets in DrugBank) we used the BANDIT voting protocol to predict specific protein targets. We required that each orphan small molecule be in at least three of BANDIT's databases, leaving us with a set of ~15,000 small molecules. To refine our initial list of predictions into a high confidence set, we required a TLR cutoff of 500, that each predicted target appear in the majority of shared target predictions, and that the highest ranked target appear in the top shared target prediction for each orphan molecule. From this list of high confidence predictions, we identified a set of small molecules predicted to bind to microtubules.

For each predicted microtubule inhibitor (MTI) we examined how it related to known MTIs using a network approach (Supplementary Fig. 9). We required that each predicted MTI have a TLR greater than 500 with at least two known MTIs. Each edge in our network represents a predicted shared target interaction with the length and width of each corresponding to the strength of the prediction (measured by the TLR value). We used the Fruchterman Reingold projection within the R igraph package. We observed a distinct clustering of known MTIs based on their mechanism of action.

Most of the novel MTIs we predicted were not easily obtained, thus we specifically focused on the subset that we could obtain from the National Cancer Institute's Developmental Therapeutics Program (Supplementary Table 2).

**Microtubule imaging/testing**. Human breast MDA-MB-231 cells were obtained from the American Type Culture Collection (ATCC, Manassas, VA) and cultured in DMEM (obtained from Corning Cellgro) with 10% fetal bovine serum and 1% penicillin and streptomycin. Cells were plated at the density of 90,000 cells/ml onto 12 mm round cover slips in 48 well plates for 24 h and then treated for 6 h with small molecules at the given concentrations. Small molecules (obtained from the NCI Drug Bank) were dissolved in DMSO and stored at $-20\,°C$. Control experiments were done using DMSO and it was <0.5% of total media volume. After 6 h drug treatment media was removed and cells were permeabilized with 0.5% Triton X-100 and fixed with PHEMO Buffer (3.7% formaldehyde, 0.05% glutaraldehyde, 0.068 M Pipes, 0.025 M HEPES, 0.015 M EGTANa$_2$, 0.003 M MgCl$_2$6H$_2$O and 10% DMSO and adjust pH = 6.8) for 10 min. Fixed cells were washed three times with PBS buffer. Cells were blocked with 10% goat serum at room temperature for 10 min. Cells were incubated with monoclonal α-tubulin antibody (1:1000 dilution, clone YL 1/2, obtained from EMD Millipore, cat#MAB1864), for 1 h and washed three times with PBS buffer before incubation with a secondary Alexa Fluor 488 goat anti-mouse antibody (1:500 dilution, obtained from Invitrogen, cat# A-11006). Cell chromatin was stained with DAPI for 5 min and washed with water three times. Cover slips were mounted and photographed in a RSM 700 microscope for microtubule visualization. DNA was counterstained with DAPI. Images were acquired with Zeiss LSM 700 confocal microscope under a 63×/1.4NA objective (Zeiss, Germany) (Fig. 3a–h, Supplementary Figs. 10–15).

A Fisher's exact test was used to determine whether the number of observed successes—defined as a predicted microtubule inhibitor showing an effect against

microtubules in imaging—was greater than what would be expected by random chance. To determine the background probability we used the number of drugs with known targets in our database that were known to target microtubules (~1%).

**Microtubule effect quantification**. Following 6 h treatment, cells (12 well plate) were washed once with warm phosphate-buffered saline. Each well was incubated with 150 μL either with low salts or high salt buffer at 37 °C for 10 min. Cell were then scraped and were either lysed in low salt buffer to test for the degree of tubulin polymerization (20 mM Tris–HCl pH 6.8, 1 mM $MgCl_2$, 2 mM EGTA, 0.5% NP-40, 1× protease inhibitor cocktail and 0.5% NP-40) or high salt buffer to test for the degree of tubulin depolymerization (0.1 M Pipes, 1 mM EGTA, 1 mM $MgSO_4$, 30% glycerol, 5% DMSO, 1 mM DTT, 0.02% NAAzide, 0.125% NP-40, 1 mM DTT and 1× protease inhibitor cocktail). Samples were spun at max speed in a tabletop centrifuge for 30 min at room temperature. The supernatant (S) was separated from the pellet (P). The pellet was resuspended in 150 μL 1 × Laemmli buffer and sonicated. Equal volumes of supernatant and pellet samples were loaded onto a 12% gel for a western blot. Tubulin bands were visualized with a DM1 monoclonal antibody (1:5000 dilution obtained from Sigma-Aldrich, cat# T9026). % Tubulin in pellet levels were calculated as the densitometric value of the pellet band divided by the total densitometric value of the pellet and supernatant bands times 100. Three biological repeats were performed (Supplementary Fig. 16).

**Treatment against resistant cell lines**. The 1A9 is a clone of the human ovarian carcinoma cell line, A2780. 1A9-ERB is a clone of the 1A9 human ovarian carcinoma cell line resistant to the effects of Eribulin mesylate. It was prepared by exposing 1A9 cells to 1 ng/ml Eribulin (obtained from Eisai pharmaceuticals) in the presence of 10 μg/ml verapamil (obtained from Acros Organics), a Pgp antagonist. The cells were maintained in the 0.5 ng/ml eribulin and 10 μg/ml verapamil concentrations. Cells were removed from this drug solution 3 days prior to any future experimentation. Additional treatment and imaging was done using the same protocols as described earlier (Supplementary Figs. 17–19).

Growth inhibition activity was determined using 96 well plates. Cells were seeded at density of 4000 cells/well for 24 h. Cell were exposed to 1 ml serial dilution (0.014–8100 nM) of different drugs for 72 h. An SRB assay was performed to determine the cell viability. $GI_{50}$ was determined as drug concentration that causes a 50% decrease in cell growth. Experiments were repeated at least three times to obtain the mean and standard deviation for each experiment.

**Characterization of ONC201-DRD2 interaction**. ONC201 dihydrochloride was obtained from Oncoceutics. GPCR arrestin recruitment and cAMP modulation reporter assays were performed using PathHunterTM (DiscoveRx) beta-arrestin cells expressing one of several GPCR targets and HitHunter cAMP cells, respectively[63]. For arrestin recruitment, cells were plated onto 384-well white solid bottom assay plates (Corning 3570) at 5000 cells per well in a 20 μL volume in the appropriate cell plating reagent. Cells were incubated at 37 °C, 5% $CO_2$ for 18–24 h. Samples were prepared in buffer containing 0.05% fatty-acid free BSA (Sigma). For agonist mode tests, samples (5 μL) were added to pre-plated cells and incubated for 90 min at 37 °C, 5% $CO_2$. For antagonist mode tests, samples (5 μL) were added to pre-plated cells and incubated for 30 min at 37 °C, 5% $CO_2$ followed by addition of EC80 agonist (5 μL) for 90 min at 37 °C, 5% $CO_2$. For Schild analysis, samples (5 μL) were added to pre-plated cells and incubated for 30 min at 37 °C, 5% $CO_2$ followed by addition of serially diluted agonist (5 μL) for 90 min at 37 °C, 5% $CO_2$. Control wells defining the maximal and minimal response for each assay mode were tested in parallel. Arrestin recruitment was measured by addition of 15 μL PathHunter Detection reagent and incubated for 1–2 h at room temperature and read on a Perkin Elmer Envision Plate Reader. For agonist and antagonist tests, data was normalized for percent efficacy using the appropriate controls and fitted to a sigmoidal dose–response (variable slope), Y = Bottom + (Top-Bottom)/(1 + 10^((LogEC50-X)*HillSlope)), where X is the log concentration of compound. For cAMP modulation, a method similar to that for arrestin recruitment was used with the DiscoveRx HitHunter cAMP XS + assay and measured by incubation with cAMP XS + Ab reagent and cAMP XS + ED/CL lysis cocktail for 1 h followed by incubation with cAMP XS + EA reagent for 3 h at room temperature.

For Schild analysis, data was normalized for percent efficacy using the appropriate controls and fitted to a Gaddum/Schild EC50 shift using global fitting, where Y = Bottom + (Top-Bottom)/(1 + 10^((LogEC-X)*HillSlope)), Antag = 1 + (B/(10^(−1*pA2)))^SchildSlope and LogEC = Log(EC50*Antag). EC50/IC50 analysis was performed in CBIS data analysis suite (Cheminnovation) and Schild analysis performed in GraphPad Prism 6.0.5.

Kinase inhibition assays for the kinome were performed by Reaction Biology Corp[30]. In vitro kinase panel profiling was performed using the "HotSpot" assay platform. Briefly, kinase, substrate, cofactors were prepared in reaction buffer at room temperature. Compound, ATP and 33P ATP was added to a final concentration of 10 μM. Spotting of the reactions was done with ion exchange filter paper. Unbound phosphate was removed in phosphoric acid. Kinase activity data was expressed as % remaining kinase activity in test samples compared to vehicle. IC50 values and curve fits were obtained.

The nuclear hormone receptor profiling (S16) was performed by DiscoverX[64–66]. In this system, a CHO cell expressing a portion of betagalactosidase in the nucleus is stably transfected with a nuclear hormone receptor construct fused to a complementary portion of beta-galactosidase prolabel. Upon ligand binding, the fusion protein translocates to the nucleus, allowing chemiluminescent detection. Cells at a range of densities were resuspended in assay buffer into white-tissue culture-treated multi-well plates and increasing concentrations of control compounds were dispensed. The plate was incubated for 3 h at 37 °C and 5% $CO_2$. PathHunter detection reagent was added to each well prior to incubation for 1 h at room temperature and reading relative luminescence. Dose response curves were plotted using Prism.

**Drug mechanism clustering**. For each drug pair we converted the TLR between them into a distance metric used to estimate closeness between any two drugs:

$$\text{BANDIT Distance Score} = \frac{1}{\text{TLR}} \qquad (3)$$

We next separated all drugs know to target microtubules that were in at least three of BANDIT's dataset. With the BANDIT distance metric as an input we created a hierarchical cluster of all known MTIs using the hclust R method with an average based clustering method. Known MTIs were labeled based on whether they were known to polymerize or depolymerize microtubules, and we observed a distinct separation based on the mechanism of action (MoA). We repeated this clustering while removing drug structures from our likelihood calculations and continued to see a MoA-based separation (Supplementary Fig. 20). This revealed that BANDIT's clustering approach is not dependent on any single data type, and that observed results are due to BANDIT's integrative approach. This analysis was then repeated using similar conditions for known protein kinases.

**Drug universe clustering**. Using the same protocol as was used to create the MTI network, we created a network of all drugs with known targets with each edge representing a predicted shared target interaction and the edge weight corresponding to the strength of the interaction. Using the KEGG drug database[67] and DrugBank[27] we annotated each drug based on its most prevalent ATC code and colored each drug accordingly. We specifically isolated out three clusters representing: (1) beta-blockers with Parkinson's medications, (2) antiretrovirals and statins and (3) opioids and microtubule inhibitors.

To get a better understanding of how orphan small molecules fit into this drug universe we computed the distance between every pair of small molecules and used multi-dimensional scaling to visualize the overall structure (Supplementary Fig. 21). We used the same distance metric as described in the mechanism of action clustering section to create a distance matrix between all small molecules (known drugs and orphan) and used the R cmdscale package for the multi-dimensional scaling. We noticed a definite structure with known drugs tightly clustering around each other, while orphan molecules had a more diffuse organization. One explanation for this structure is that drugs with known targets are more likely to be used to treat patients and thus may have similar effects due to safety precautions, whereas orphan molecules which have not gone through clinical trials and FDA approval are more likely to have a wide variety of effects and characteristics.

**Reporting summary**. Further information on research design is available in the Nature Research Reporting Summary linked to this article.

## Data availability

All data used for the analyses in this paper is publicly available. Further information on the specific sources could be found in the "Datasets" portion of the Methods section.

## Code availability

Select pieces of custom code can be made available upon request.

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

## Acknowledgements

The authors would like to thank the members of the Elemento Lab and Giannakakou Lab for their feedback and discussions. O.E. is supported by NIH grants U24 CA210989, R01 CA194547, P50CA211024, UL1TR002384 and Emerson Collective Cancer Research Fund. Support was also provided for N.S.M. and K.G. by the PhRMA Foundation Pre Doctoral Informatics Fellowship and by the Tri-Institutional Training Program in Computational Biology and Medicine.

## Author contributions

N.S.M. and O.E. conceived, designed, and developed the methodology for this work. N.S.M, L.H., K.G. and O.E. analyzed and interpreted the data and developed the computational methodology. P.K.K., G.G. and P.G. designed and carried out the experimental screening of the anti-microtubule compounds. M.S. and J.E.A. designed and supervised all experimental work related to ONC201. O.E. supervised the study.

## Competing interests

O.E. and N.S.M are co-founders and equity holders in One Three Biotech, Inc, a company that may use some of the methods described in this article for commercial purpose. O.E. and N.S.M. have filed a patent application on the Bayesian data integrative computational method for predicting binding targets of chemicals (BANDIT) described in this article (US Patent App. 16/315,625). J.E.A. and M.S. are employees and shareholders of Oncoceutics. The remaining authors declare no competing interests.
