## [Peer Review File · Nature Communications]

Reviewers' comments:

Reviewer #1 (Remarks to the Author):

The authors proposed a Bayesian machine-learning approach, BANDIT, to support drug- target identification.

The topic of the paper is of great interest since novel computational methodologies are required to speed up the drug screening of novel potential drug candidates.

In addition, the method leverage on the integration of multiple data sources on drug information.

The authors validated the novel predicted drug-target interactions with experimental validation as well as with existing data. This increase the effectiveness of their approach.

I have some comments on the methodology to well understand how the integration of multiple data sources effectively improves the resulting predictions.

- In fig.2A the authors plotted AUCs of the method by considering an increasing number of data sources. The figure shows how BANDIT performances increase by integrating more data sources. However, they plot the different AUCs starting from the less informative data source to the most informative one, so this would may be expected. In addition, Figure S3 shows AUC of each distinct data source confirming single AUC values lower than the ones obtained with the integrated model. I would like to see if this happens also by plotting AUCs data sources starting from the most informative data source and adding the subsequent ones. For example, if considering the 2 most informative data sources the authors will find a comparable AUC, they should provide how the results will change. I would like also to see PCA of all the data integrated.

Minor:

- Figure are not cited with a correct numerical order.
- figure S1 is not immediate to understand. The authors should make clearer the idea of different scores.
- Figure S4 has some weird symbols
- Authors need to revise English language and editing issues

Reviewer #2 (Remarks to the Author):

General comments:

The authors propose a method to infer drug-target interactions in the Bayesian framework. Predicting unknown targets of drugs is an important issue in drug discovery. However, there is no conceptual advance in this paper. There is no fair performance comparison with similar previous works with the same objective. The performance evaluation procedure is problematic.

Specific major comments:

The title is too general. The use of biological big data is common in the research of drug target prediction. The title should contain specific points of the paper.

As a new prediction, the proposed method (BANDIT) predicted dopamine receptor 2 as a target of ONC201. The previous computational methods failed to make the prediction?

There are many scientific papers reporting that dopamine receptor 2 as a target of ONC201. Examples include the following papers.

Neoplasia. 2018 Jan;20(1):80-91
Neoplasia. 2018 Jan; 20(1): 80–91.
Oncotarget. 2017 Oct 3; 8(45): 79298–79304.
I'm afraid that this prediction is not new.

What is the mechanism of anti-cancer effect regarding dopamine receptor 2?
I'm afraid that dopamine receptor 2 is not a primary target but just a off-target of ONC201.

There are many important previous works on drug-target interaction prediction or compound-protein interaction prediction with machine learning methods. However, the authors did not cite/discuss the previous works in the manuscript. Examples include the following papers:
Faulon et al., Genome scale enzyme-metabolite and drug-target interaction predictions using the signature molecular descriptor. *Bioinformatics*, 24:225–233, 2008.
Jacob et al., Protein-ligand interaction prediction: an improved chemogenomics approach. *Bioinformatics*, 24:2149–2156, 2008.
Bleakley, et al, Supervised prediction of drug-target interactions using bipartite local models, *Bioinformatics*, Vol.25, pp.2397-2403, 2009.
Yabuuchi et al, Analysis of multiple compound-protein interactions reveals novel bioactive molecules. *Mol Syst Biol* 2011, 7:472.
van Laarhoven,T. et al. (2011) Gaussian interaction profile kernels for predicting drug-target interaction. *Bioinformatics*, 27, 3036–3043.
Shi et al, Protein-chemical Interaction Prediction via Kernelized Sparse Learning SVM, *Pacific Symposium on Biocomputing* 18:41-52(2013)
Wang et al, Drug Target Predictions Based on Heterogeneous Graph Inference, *Pacific Symposium on Biocomputing* 18:53-64(2013)
Yamanishi et al (2014), DINIES: drug-target interaction network inference engine based on supervised analysis, *Nucleic Acids Research*, 42, W39-W45.
Deng et al (2018), DrugE-Rank: Predicting Drug-Target Interactions by Learning to Rank, *Methods Mol Biol*.

The authors should discuss the relationship with the previous works on the same topic. They should also compare the performance with the existing methods. Otherwise, it is impossible to evaluate the significance of the proposed method.

They cited the following papers, but they did not compare the performance with them on a large scale.
Campillos et al., Drug target identification using side-effect similarity. *Science*, 321(5886):263–266, 2008.
Keiser,M. et al., Predicting new molecular targets for known drugs. *Nature*, 462, 175–81, 2009.
Wang et al., Prediction of drug-target interactions for drug repositioning only based on genomic expression similarity. *PLoS Comput Biol*. 2013.

The authors claim that a strength of the Bayesian framework is that it can easily accommodate new features. However, there are many machine learning methods that can easily accommodate new features. For example, kernel methods can integrate many heterogeneous data/features.

The authors claim that DICE similarity is better than other similarity. I do not understand why.

The authors used AUROC as a performance measure, but it is not appropriate when the number of positives is much lower than the number of negatives. The AUROC score tends to be higher as the number of negatives is much larger than the number of positives. How did you handle the problem of the positive/negative ratio in the cross-validation? The use of Precision-Recall curve or MCC score would be more appropriate than ROC curve.

One concern is that the high accuracy scores are just due to homologous relationships between

similar drugs or between similar proteins. The authors should evaluate the generalization properties of the method after removing very similar drugs (e.g., derivatives from the same compound) and proteins from the dataset.

To test the accuracy, the authors used leave-one-out cross validation. It is not appropriate, because the resulting prediction accuracy is over-estimated. 3-fold or 5-fold cross-validation would be more appropriate.

Minor comments:

The following paper in Reference should be fixed.

48 J. Ishizawa et al.

Reviewer #3 (Remarks to the Author):

The manuscript "A new big-data paradigm for target identification and drug discovery" by Madhukar et al is a well-written article describing a novel paradigm for assisting in the determination of target identification and mechanism of action of bioactive small molecules. The integrated and multi-faceted nature of the method takes into account methodologies that are already commonly in use to have an immediate impact along with the versatility to incorporate new types of datasets to remain relevant over time. Discussion of the relative ability of different features to differentiate drug classes alone is a valuable contribution to the literature.

The detailed description of the importance of microtubule targeting agents in the clinic, the differences among these compounds, and the need for identification of novel agents of this class that circumvent clinically relevant forms of drug resistance is an important introduction into the relevance of the use of BANDIT to identify new microtubule targeting agents from a library of orphan small molecules. While the analysis of the biological effects of the 24 compounds predicted to have microtubule targeting activity is well thought out and described, there are a few suggestions to make this section more robust and to increase its impact.

Is there any idea why about half of the compounds that were hits from the BANDIT analysis did not appear to have microtubule stabilizing or destabilizing activity in cells? Is there a possibility that these compounds are not able to target microtubules in the cell-based assay due to poor cellular permeability? If so, it might be interesting to determine whether these compounds can directly interact with purified tubulin to effect polymerization in biochemical assays. Otherwise, it may be interesting to revisit whether there were aspects of the BANDIT algorithm that were similar in the negative hits that may be worth exploring as ways to further refine the prediction. While a 54% success rate is indeed much higher than chance, it is worth the effort to determine whether anything more can be learned about BANDIT after validation of the potential hits (i.e. is there a similar structural feature in several of the negative hits?)

It appears that this is the first paper to describe the eribulin-resistant 1A9 cell line used in this manuscript. Therefore, some additional description of the mechanism of resistance of these cells would be valuable. Although the lines were generated in the presence of verapamil, is P-glycoprotein upregulated in these cells? Have any tubulin mutations been identified that would correspond to eribulin (or other MTA) binding? From supplemental figure 16 and table 3 it appears that this line is much less resistant to colchicine, which is worth mentioning in the text. It would be valuable to note whether the structure of each of these compounds is consistent with potential colchicine-site binding. Ideally, colchicine displacement assays could be easily performed to determine if each of these molecules bound at that site. This is an important point since there are a large number of colchicine-site small molecules that circumvent clinically relevant mechanisms of drug resistance to MTAs that are in clinical and preclinical development. If these compounds

circumvent drug-resistance by binding independently of the colchicine site, it may increase their potential clinical impact and if the current findings have identified a novel class of microtubule destabilizer, it may be worth indicating that this is being followed up on in subsequent work to signify the value of this analysis.

Line 274 mentions the “antitumor activity” of the cytotoxicity assay. This assay employs cell lines, not tumors and therefore the results should not be characterized as antitumor unless in vivo tumors have been assayed.

There is some confusion whether table S3 demonstrates cytotoxicity or growth inhibition. These are distinct measurements. The methods section, line 651, indicates that the IC50 calculated is based on the concentration required to inhibit cell growth by 50%. This would be consistent with the GI50 value described in dataset #1 of the methods, line 441, used by the NCI. If this is the case, the term cytotoxic/cytotoxicity in lines 266, 275, 281, and 648 should be changed.

Line 282 says that compound 15 was “found to almost completely reverse drug-resistance”. It would be more appropriate to say that it circumvented drug resistance since there is no indication that it reversed the drug-resistance mechanism of that cell line, only that it was able to retain efficacy in spite of it.

Line 306, remove “a” before dopaminergic

Lines 322 & 325 – change uM to μ M

In the methods section it indicates that a GI50 measurement was used, therefore, in line 440 “50% decrease in cells” should be changed to “50% decrease in cell growth”.

There is a formatting problem in figure S4

We are very grateful to the reviewers for their insightful comments. We believe that we have addressed all of them in details and have as a result substantially improved our manuscript. Below please find a point-by-point response to each comment, explaining how we addressed each one of them.

Reviewer 1:

Comment: In fig.2A the authors plotted AUCs of the method by considering an increasing number of data sources. The figure shows how BANDIT performances increase by integrating more data sources. However, they plot the different AUCs starting from the less informative data source to the most informative one, so this would may be expected. In addition, Figure S3 shows AUC of each distinct data source confirming single AUC values lower than the ones obtained with the integrated model. I would like to see if this happens also by plotting AUCs data sources starting from the most informative data source and adding the subsequent ones. For example, if considering the 2 most informative data sources the authors will find a comparable AUC, they should provide how the results will change. I would like also to see PCA of all the data integrated.

Revision: This was something we were interested in as well when we were creating the final model. In order to best understand the relationship addition order had on the overall accuracy, we measured the average AUC at different numbers of data types and different additional orders (Table S1), and also observed a steady increase across the board. However to address this particular comment, we have included ROC curves (Figure R1) where we added in data types from most individually predictive to least individually predictive (measured by single data type AUC). Here we also observed a steady increase in total AUC as new data types were added.

Figure R1

Comment:

- Figure are not cited with a correct numerical order.

- figure S1 is not immediate to understand. The authors should make clearer the idea of different scores.

- Figure S4 has some weird symbols
- Authors need to revise English language and editing issues

Revision: All suggested editorial comments and figure revisions shown above have been addressed in the manuscript and figures.

Reviewer 2:

Comment: As a new prediction, the proposed method (BANDIT) predicted dopamine receptor 2 as a target of ONC201. The previous computational methods failed to make the prediction?

Revision: Yes, in order to test the power of BANDIT vs other methods, we tested ONC201's target prediction using two methods – SEA and SuperPred – neither of which predicted DRD2 as a top binding target. We have updated the manuscript with reference to the exact methods that were used.

Comment: There are many scientific papers reporting that dopamine receptor 2 as a target of ONC201. Examples include the following papers: Neoplasia. 2018 Jan;20(1):80-91; Neoplasia. 2018 Jan; 20(1):80–91; Oncotarget. 2017 Oct 3; 8(45): 79298–79304. I'm afraid that this prediction is not new.

Revision: In fact the articles the reviewer listed were published by our team and actually built off of the original work in this paper. For instance in the Neoplasia article they directly cite “Madhukar, NS, Khade, PK, Huang, L, Gayvert, K, Galletti, G, Stogniew, M, Allen, JE, Giannakakou, P, and Elemento, O. A new big-data paradigm for target identification and drug discovery. bioRxiv, ; 2017,” which is the biorxiv preprint of this same publication.

Additionally the earliest (2017) Oncotarget article (“Discovery and clinical introduction of first-in-class imipridone ONC201”) states that the target of ONC201 was originally discovered using the BANDIT computational method described in the current manuscript.

To summarize, the current manuscript is the first to thoroughly describe the discovery and validation of DRD2 as ONC201's target. The 3 papers cited by the reviewer simply refer to the preprint version of the current manuscript and do not describe the discovery or validation of DRD2 as ONC201's target.

Comment: What is the mechanism of anti-cancer effect regarding dopamine receptor 2? I'm afraid that dopamine receptor 2 is not a primary target but just a off-target of ONC201

Revision: Thank you for this comment. Since our initial finding of ONC201's binding to DRD2, we have worked with researchers at Oncoceutics and other institutes to more fully explore the anti-cancer mechanism in relation to DRD2 targeting. We have recently published some of these results in: Neoplasia. 2018 Jan;20(1):80-91. In particular I would direct the reviewer's attention to Figure 2B of the Neoplasia article where we show that over expression of DRD2 increases apoptosis induced by ONC201 and Figure 3 that shows how DRD2 knockdown can phenocopy ONC201's signaling and affect its activity. Additionally we've seen how dysregulation of dopamine receptor expression can correlate with resistance to ONC201 and how expression of dopamine receptors can be used as a biomarker for ONC201 activity in gliomas (and this biomarker is currently being used in commercial development). The most recent abstract for this finding was at SNO last year:

https://academic.oup.com/neuro-oncology/article-abstract/19/suppl_6/vi60/4590813?redirectedFrom=fulltext

Overall these results support the hypothesis that DRD2 is ONC201's primary target and directly impacts the observed anti-cancer effects.

Comment: The authors should discuss the relationship with the previous works on the same topic. They should also compare the performance with the existing methods. Otherwise, it is impossible to evaluate the significance of the proposed method.

Revision: Thank you for your comment and we agree that there are multiple machine learning methods that are able to handle diverse data types. To better address some of the benefits of the Bayesian approach we have updated the manuscript throughout the main text and supplement with this information. For instance, some of the benefits that led to our choice of a Bayesian network were:

1. Computational power – we observed a much quicker training time when using a Bayesian framework compared to other machine learning approaches
2. Interpretability – Since predictive power is measured in terms of odds ratio and probabilities we found it was easier to understand what individual data types led to a specific prediction.
3. New Data – New data types could be added in without needing to retrain the entire model for the other data types.

Comment: The authors claim that DICE similarity is better than other similarity. I do not understand why.

Revision: We tested DICE similarity vs other similarity types and evaluated performance based on how each similarity metric was able to separate shared-target vs non-shared-target drugs (measured by KS-test). DICE similarity outperformed other methods (such as Tanimoto similarities) based on this metric. We have updated the manuscript supplement to explain this further.

Comment: The authors used AUROC as a performance measure, but it is not appropriate when the number of positives is much lower than the number of negatives. The AUROC score tends to be higher as the number of negatives is much larger than the number of positives. How did you handle the problem of the positive/negative ratio in the cross-validation? The use of Precision-Recall curve or MCC score would be more appropriate than ROC curve.

Revision: For cross validation we sub-sampled the two classes (ST and non-ST drug pairs) and required the ratio of true positives (ST pairs) to true negatives (non-ST pairs) to remain the same as the total set. However to further measure accuracy we measured the TP/FP ratio at different likelihood value cutoffs (Figure 2). This method of evaluating Bayesian likelihoods for biological values when there is a significant difference in the number of positives and negatives was previously established by Jansen et al, Science, "A Bayesian Networks Approach for Predicting Protein-Protein Interactions from Genomic Data," 2003. To further evaluate the predictive accuracy we have also included a supplementary figure (Figure S3-4) comparing the AUROC and AUPRC for our final method compared to randomly shuffling likelihood values (thereby representing random likelihoods). In both comparisons we found that BANDIT significantly outperformed random selection.

Comment: One concern is that the high accuracy scores are just due to homologous relationships between similar drugs or between similar proteins. The authors should evaluate the generalization properties of the method after removing very similar drugs (e.g., derivatives from the same compound) and proteins from the dataset.

Revision: Thank you for this comment. This was in fact one of initial concerns when developing the model. To test for it, we measured the number of drug pairs which had a high degree of structural similarity. We found that out of all drug pairs we evaluated, less than .05% had a structural similarity value higher than .8, with less than .01% having a similarity value higher than .9. Based on this information we do not suspect that our true positive rate was simply due to homologous drugs, given that they represent such a small portion of the overall tests.

Comment: To test the accuracy, the authors used leave-one-out cross validation. It is not appropriate, because the resulting prediction accuracy is over-estimated. 3-fold or 5-fold cross-validation would be more appropriate.

Revision: We redid our initial calculation using 5-fold cross validation and saw little difference in overall result (Figure R2):

Figure R2

We have updated the manuscript to include this observation in relation to leave-one-out cross validation.

Comment: The following paper in Reference should be fixed.
48 J. Ishizawa et al.

Revision: we fixed this issue.

Reviewer 3:

Comment: Is there any idea why about half of the compounds that were hits from the BANDIT analysis did not appear to have microtubule stabilizing or destabilizing activity in cells? Is there a possibility that these compounds are not able to target microtubules in the cell-based assay due to poor cellular permeability? If so, it might be interesting to determine whether these compounds can directly interact with purified tubulin to effect polymerization in biochemical assays. Otherwise, it may be interesting to revisit whether there were aspects of the BANDIT algorithm that were similar in the negative hits that may be worth exploring as ways to further refine the prediction. While a 54% success rate is indeed much higher than chance, it is worth the effort to determine whether anything more can be learned about BANDIT after validation of the potential hits (i.e. is there a similar structural feature in several of the negative hits?)

Revision: Thank you for your comment. This is actually very interesting and in agreement with the known discrepancies reported for small molecules targeting purified microtubules but not microtubules in cells. Such discrepancies are usually attributed to poor cellular permeability, as the reviewer points out. To address this point we had initially performed a “crude-tubulin” assay in which cell membranes were permeabilized with detergent in the presence of 10 uM of each compound for 30 min at 37°C, followed by centrifugation to separate polymerized from soluble tubulin. Under these conditions, the plasma membranes were permeabilized, thereby avoiding any cell permeability issues, and the compounds were incubated directly with crude-tubulin extracts for a short period of time. This assay identified 14/24 compounds as having anti-tubulin

activity; 13/24 compounds were identical to those identified by the cell-based tubulin polymerization assay (6 hr of treatment followed by cell lysis) and the tubulin immunofluorescence assay. Compound #3 was identified by the crude-tubule assay as an additional positive hit, increasing the success rate to 58%. We have modified the manuscript to describe these results.

The high concordance of the crude-tubulin and cellular tubulin assays together with the small number of available compounds suggested that an *in vitro* tubulin binding assay may not add much to BANDIT at this point. However, in the future as we get more compounds predicted to target tubulin we will consider testing them both in vitro and in cells to further refine the algorithm.

Further examining the compound structures we did not observe any immediate patterns within the negative and positive hits. At this point, there are too few tested compounds to broadly generalize, but we expect that a clearer pattern may emerge as more compounds predicted by BANDIT are tested experimentally. However we did notice that the negative hits had much lower TLR scores than the positive hits (median TLR for the top ST prediction was 173 for the negative hits vs 24,000+ for the positive hits). This may indicate that in the future a higher TLR cutoff should be applied to select top compounds.

Comment: Although the lines were generated in the presence of verapamil, is P-glycoprotein upregulated in these cells? Have any tubulin mutations been identified that would correspond to eribulin (or other MTA) binding? From supplemental figure 16 and table 3 it appears that this line is much less resistant to colchicine, which is worth mentioning in the text. It would be valuable to note whether the structure of each of these compounds is consistent with potential colchicine-site binding. Ideally, colchicine displacement assays could be easily performed to determine if each of these molecules bound at that site. This is an important point since there are a large number of colchicine-site small molecules that circumvent clinically relevant mechanisms of drug resistance to MTAs that are in clinical and preclinical development. If these compounds circumvent drug-resistance by binding independently of the colchicine site, it may increase their potential clinical impact and if the current findings have identified a novel class of microtubule destabilizer, it may be worth indicating that this is being followed up on in subsequent work to signify the value of this analysis.

Revision: The reviewer correctly observed that this is the first manuscript which reporting on the Eribulin resistant ovarian cancer cell line 1A9-ERB. We are currently investigating the potential mechanism(s) responsible for the resistance to Eribulin, and these data will be part of another manuscript. In terms of P-gp expression, although P-gp was detectable in 1A9-ERB cells, the cell line retained profound resistance to Eribulin (470 fold-resistance to Eribulin as compared to parental 1A9 cells) in the presence of the P-gp inhibitor verapamil. These data clearly suggested that additional intracellular mechanisms of resistance were present. Compound #15 was the most potent and its potency was not enhanced in the presence of verapamil, suggesting that compound #15 was active despite P-gp expression and despite the presence of additional intracellular mechanisms of resistance.

With regards to the presence of tubulin mutations, we have performed both RNA-sequencing and direct Sanger Sequencing of the different beta-tubulin and alpha tubulin isotypes expressed in these cell lines and did not identify any tubulin mutations. Additionally, we have assessed potential differences in total tubulin content or tubulin post-translational modifications in the ERB-resistant vs parental 1A9 cells and did not detect any significant differences.

With regards to colchicine, we agree with the reviewer that colchicine had a lower GI50 against the resistant cells (560 nM) compared to the GI50 of Eribulin (2397 nM), but colchicine was less potent than Vinblastine (GI50: 208 nM). The degree of resistance is driven by the GI50 of each drug against the resistant and against the parental 1A9 cells. Colchicine has a much higher GI50 against the parental cells (1-2 logs higher than the IC50s of Eribulin and vinblastine respectively). Since these differences in the parental line cannot be directly attributed to potential differences in tubulin binding, colchicine displacement assays are not particularly relevant here.

Comment: Line 274 mentions the “antitumor activity” of the cytotoxicity assay. This assay employs cell lines, not tumors and therefore the results should not be characterized as antitumor unless in vivo tumors have been assayed.

Revision: Thank you for identifying this misnomer, we have corrected the manuscript accordingly.

Comment: There is some confusion whether table S3 demonstrates cytotoxicity or growth inhibition. These are distinct measurements. The methods section, line 651, indicates that the IC50 calculated is based on the concentration required to inhibit cell growth by 50%. This would be consistent with the GI50 value described in dataset #1 of the methods, line 441, used by the NCI. If this is the case, the term cytotoxic/cytotoxicity in lines 266, 275, 281, and 648 should be changed.

Revision: As the reviewer correctly points out, Table S3 shows growth inhibition data, hence the correct term should be GI50 and not IC50. We have updated the manuscript accordingly.

Comment: Line 282 says that compound 15 was “found to almost completely reverse drug-resistance”. It would be more appropriate to say that it circumvented drug resistance since there is no indication that it reversed the drug-resistance mechanism of that cell line, only that it was able to retain efficacy in spite of it.

Revision: Thank you for catching this. We have modified the manuscript accordingly.

Comment:

- Line 306, remove “a” before dopaminergic
- Lines 322 & 325 – change uM to μ M
- In the methods section it indicates that a GI50 measurement was used, therefore, in line 440 “50% decrease in cells” should be changed to “50% decrease in cell growth”.
- There is a formatting problem in figure S4

Revision: All suggested editorial comments shown above have been addressed in the manuscript

Reviewers' comments:

Reviewer #1 (Remarks to the Author):

The authors well addressed all the answers to the reviewers.
The paper is now more clear and complete.

Reviewer #2 (Remarks to the Author):

General comments:

The authors did not address all the points raised by reviewers. They ignored several comments of reviewers.

The authors should have highlighted the changed parts in the revised manuscript. It is very difficult to identify the revised points.

Specific major comments:

In the response letter, the authors did not address the following comments in the previous review:

The title is too general. The use of biological big data is common in the research of drug target prediction. The title should contain specific points of the paper.

The title of this paper should be modified. Big data were used in most previous works on drug target prediction with machine learning methods.

"A Bayesian approach for drug target identification with multiple data sources" would be better than "A New Big-Data Paradigm for Target Identification and Drug Discovery".

The same authors already published several papers that reported dopamine receptor 2 was a target of ONC201.

Neoplasia. 2018 Jan; 20(1): 80-91

Oncotarget. 2017 Oct 3; 8(45): 79298–79304.

The authors did not cite the Oncotarget paper in this paper.

It is problematic.

In the Introduction section, the authors say "Recently we have seen more methods emerging that integrate multiple different data types within a data-driven or machine-learning framework [18-25]". However, it is not correct. Some methods of cited papers [18-25] are based on single data of drugs and single data of proteins. Please check the details.

The authors insist that the proposed method has the following advantages over the previous methods.

1. Computational power – we observed a much quicker training time when using a bayesian framework compared to other machine learning approaches
2. Interpretability – Since predictive power is measured in terms of odds ratio and probabilities we found it was easier to understand what individual data types led to a specific prediction.
3. New Data – New data types could be added in without needing to retrain the entire model for the other data types.

However, kernel-based methods in the previous works have the above features.

For example, SITAR is computationally very efficient (Journal of computational biology, 18, 133-

145, 2011).

In the response letter, the authors did not address the following comments in the previous review:

They cited the following papers, but they did not compare the performance with them on a large scale.

Campillos et al., Drug target identification using side-effect similarity. *Science*, 321(5886):263–266, 2008.

Keiser, M. et al., Predicting new molecular targets for known drugs. *Nature*, 462, 175–81, 2009.

Wang et al., Prediction of drug-target interactions for drug repositioning only based on genomic expression similarity. *PLoS Comput Biol*. 2013.

In the response letter, the authors did not address the following comments in the previous review:

The authors claim that a strength of the Bayesian framework is that it can easily accommodate new features. However, there are many machine learning methods that can easily accommodate new features. For example, kernel methods can integrate many heterogeneous data/features.

The authors said that DICE similarity outperformed other methods (such as Tanimoto similarities) and updated the manuscript supplement to explain this further. However, I could not find the corresponding parts. Where is the comparison result of DICE and Tanimoto? Figure S1 is just a figure of similar and dissimilar chemical structures. What is the mathematical definition of the DICE similarity?

The authors included a supplementary figure (Figure S3-4) comparing the AUROC and AUPRC for the proposed method and randomly shuffling likelihood values and showed that BANDIT significantly outperformed random selection. It is not surprising that the proposed method works better than random inference. The authors should show that the proposed method works better than the previous methods.

For the dependency of accuracy on homologous relationships between drugs or between proteins: The authors just showed statistics of structural similarity value. The authors should show the accuracy after removing very similar drugs and proteins from the dataset.

Reviewer #3 (Remarks to the Author):

A New Big-Data Paradigm for Target Identification and Drug Discovery is a very well-thought out and well-written manuscript describing a novel paradigm for assisting in the determination of target identification and mechanism of action of bioactive small molecules. The revised manuscript is improved on every front by thoughtfully addressing the reviewer's comments. This manuscript describes a robust methodology to analyze the relative value of different data-sets on determining the mechanism of action of small molecules and how to leverage these diverse parameters (with the ability to incorporate additional measurements as they become more widely available) to gain better prediction power. The successful application of this model in no less than 3 robust analyses highlights its versatility. I highly recommend this manuscript for acceptance in *Nature Communications*.

With a fresh reading of the improved manuscript, I do have one comment that does not need to be incorporated into the current manuscript. It is clear that the NCI60 results are one of the more

informative data-sets used in the BANDIT analysis. However, it appears that only the growth inhibitory potency of the compounds in the 60 cell lines is used. Would the addition of relative maximal efficacy among the 60 cell lines be an additional parameter that could even further improve the predictive power of the analysis? Since this data is already being used for potency, it seems like an easy way to add an additional parameter of cytotoxic efficacy (either Emax or LC50), which may be at least, if not more valuable than just using these data to measure growth inhibition.

Minor comments:

Table 3 in supplemental information needs to be additionally revisited for a few discrepancies:

- Vinblastine is misspelt
- The methods section clearly describes GI50 calculations of antiproliferative effects, but IC50 is presented in the table and cytotoxicity is described in the table legend
- The table 3 legend describes reversing resistance instead of circumventing resistance

MDA-MD-231 in the methods section (line 597) should be corrected to MDA-MB-231

Comment 1: The title is too general. The use of biological big data is common in the research of drug target prediction. The title should contain specific points of the paper. The title of this paper should be modified. Big data were used in most previous works on drug target prediction with machine learning methods.

"A Bayesian approach for drug target identification with multiple data sources" would be better than "A New Big-Data Paradigm for Target Identification and Drug Discovery".

Response to Comment 1: We agree to change the title to "A Bayesian machine learning approach for drug target identification using diverse data types"

Comment 2: The same authors already published several papers that reported dopamine receptor 2 was a target of ONC201. Neoplasia. 2018 Jan;20(1):80-91 Oncotarget. 2017 Oct 3; 8(45): 79298–79304. The authors did not cite the Oncotarget paper in this paper. It is problematic.

Response to Comment 2: Thank you for your comment. The Oncotarget was published while this manuscript was under review. We have updated the manuscript to include this reference. We note again that the Oncotarget paper referenced the preprint of the present manuscript for discovery of DRD2 as the ONC201 target. In another recent paper also referencing the preprint, we found that Dopamine Receptor D5 (DRD5) is a Modulator of Tumor Response to DRD2 antagonism by ONC201 <https://www.ncbi.nlm.nih.gov/pubmed/30559168>. We also cited this new paper in the newly revised manuscript.

Comment 3: In the Introduction section, the authors say "Recently we have seen more methods emerging that integrate multiple different data types within a data-driven or machine-learning framework [18-25]". However, it is not correct. Some methods of cited papers [18-25] are based on single data of drugs and single data of proteins. Please check the details.

Response to Comment 3: Thank you for the comment. We have updated the introduction to better reflect the underlying methodologies of previous methods. Additionally we have expanded the introduction to better reflect the differences in data integration for past methods.

Comment 4: The authors insist that the proposed method has the following advantages over the previous methods.

1. Computational power – we observed a much quicker training time when using a bayesian framework compared to other machine learning approaches

2. Interpretability – Since predictive power is measured in terms of odds ratio and probabilities we found it was easier to understand what individual data types led to a specific prediction.

3. New Data – New data types could be added in without needing to retrain the entire model for the other data types.

However, kernel-based methods in the previous works have the above features.

For example, SITAR is computationally very efficient (Journal of computational biology, 18, 133-145, 2011).

Response to comment 4: We do agree that there are multiple different approaches that may be used to approach drug target prediction, however one aspect of the Bayesian approach that we believe gives it an advantage to other approaches (including kernel based approaches) is the feature level interpretability. Often times in standard kernel based methods is it difficult to understand what exactly the method has learned or how an individual prediction is driven by different features. Since Bayesian approaches are driven by probability distributions it is much more intuitive to understand what the method is learning. Additionally since the final likelihood value is a product of the individual likelihood ratios it is easy to determine which specific similarity (structural, genomic, efficacy, etc) led to a prediction of two drugs sharing a target. We also note that unlike many computational approaches, our approach is supported by substantial experimental data on ONC201 and novel microtubule inhibitors.

Comment 5: In the response letter, the authors did not address the following comments in the previous review: They cited the following papers, but they did not compare the performance with them on a large scale. Campillos et al., Drug target identification using side-effect similarity. Science, 321(5886):263–266, 2008. Keiser, M. et al., Predicting new molecular targets for known drugs. Nature, 462, 175–81, 2009. Wang et al., Prediction of drug-target interactions for drug repositioning only based on genomic expression similarity. PLoS Comput Biol. 2013.

Response to Comment 5: Thank you for your comment. We have modified the manuscript to include a comparison (lines 493-506) to Campillos et al.'s method (where we were able to easily obtain their raw predictions). We found that when we compared the predictions with the highest scores across both methods BANDIT was better able to pinpoint known shared-target drugs (76% accuracy vs Campillos's 47%).

Furthermore, in Wang et al. they report an AUROC of .66 based on their method's (BAES) capability to differentiate drug pairs (from DrugBank) that are known to share at least one target from those which share none. In a similar test, we found that BANDIT produced an AUROC of $\sim .90$, a clear sign of improved performance compared to the BAES method.

Comment 6: In the response letter, the authors did not address the following comments in the previous review: The authors claim that a strength of the Bayesian framework is that it can easily accommodate new features. However, there are many machine learning methods that can easily accommodate new features. For example, kernel methods can integrate many heterogeneous data/features.

Response to Comment 6: Thank you for your comment. While there are in fact many methods that can accommodate new features, one pro for BANDIT's Bayesian approach is that new features can be integrated without retraining the entire method. Since each feature is modeled independently, the likelihoods and probabilities for new features can be calculated without having to change the prior calculated likelihoods of existing features. This, in addition to other benefits (such as interpretability) led us to choose a Bayesian approach.

Comment 7: The authors said that DICE similarity outperformed other methods (such as Tanimoto similarities) and updated the manuscript supplement to explain this further. However, I could not find the corresponding parts. Where is the comparison result of DICE and Tanimoto? Figure S1 is just a figure of similar and dissimilar chemical structures. What is the mathematical definition of the DICE similarity?

Response to Comment 7: To compare the Tanimoto and DICE coefficient based similarities, we computed the D statistic from a KS-test comparing shared target and non-shared target drug pairs for each similarity type. We found that the DICE similarity values were better able to separate out shared target drugs than Tanimoto similarity ($D_{\text{DICE}} = .39$ vs. $D_{\text{TANI}} = \sim .30$). Additionally looking at other reports, in "Mining the Biomedical Literature to predict shared drug targets in DrugBank," Caniza et al. (2017) also

found that the Tanimoto similarity value on its own was not able to separate out drug pairs that shared targets vs those that shared none.

Comment 8: The authors included a supplementary figure (Figure S3-4) comparing the AUROC and AUPRC for the proposed method and randomly shuffling likelihood values and showed that BANDIT significantly outperformed random selection. It is not surprising that the proposed method works better than random inference. The authors should show that the proposed method works better than the previous methods.

Response to comment 8: We have updated the manuscript to include evidence of BANDIT's increased accuracy compared to the method from Campillos et al. Additionally, Wang et al. reported an AUROC of .66 based on their method's (BAES) capability to differentiate drug pairs (from DrugBank) that are known to share at least one target from those which share none. In a similar test, we found that BANDIT produced an AUROC of ~.9, a clear sign of improved performance compared to previous methods.

Comment 9: For the dependency of accuracy on homologous relationships between drugs or between proteins:

The authors just showed statistics of structural similarity value. The authors should show the accuracy after removing very similar drugs and proteins from the dataset.

Response to Comment 9: Thank you for your comment. Since we did not use protein based similarities in our calculations we focused our revision analysis on structurally similar (homologous) drug pairs. To further test our assertion that BANDIT's accuracy was not being driven by homologous drugs, we removed all drugs with a structural similarity of $> .9$ to another tested drug and then recalculated the final AUC with all 5 included data types. We found that even after removing these structurally similar drugs there was only a modest drop in the total AUC (decrease of .0105). Additionally, when lowering the structural similarity threshold to .85 we observed that the total AUC only decreased by .014. Overall, this shows how BANDIT maintains a high accuracy even after the removal of homologous drugs.

Comment 10: Table 3 in supplemental information needs to be additionally revisited for a few discrepancies:

- Vinblastine is misspelt
- The methods section clearly describes GI50 calculations of antiproliferative effects, but IC50 is presented in the table and cytotoxicity is described in the table legend
- The table 3 legend describes reversing resistance instead of circumventing resistance
- MDA-MD-231 in the methods section (line 597) should be corrected to MDA-MB-231

Response to comment 10: Thank you for your comment. We have corrected these typos in the final manuscript and supplement.

Reviewers' comments:

Reviewer #1 (Remarks to the Author):

The authors well addressed the questions excepts the ones on the comparison with previous methods.

A clear section with results on the comparisons with all the methods (Campillos et al, wiser et al, Wang et al) using the data the authors included in BANDIT if possible. A table with the resulting performances would add robustness to the method and the paper.

Reviewer #2 (Remarks to the Author):

Specific major comments:

> Comment 2: The same authors already published several papers that reported dopamine receptor 2 was a target of ONC201. *Neoplasia*. 2018 Jan; 20(1): 80-91. *Oncotarget*. 2017 Oct 3; 8(45): 79298–79304. The authors did not cite the *Oncotarget* paper in this paper. It is problematic.

>

> Response to Comment 2: Thank you for your comment. The *Oncotarget* was published while this manuscript was under review. We have updated the manuscript to include this reference. We note again

that the *Oncotarget* paper referenced the preprint of the present manuscript for discovery of DRD2 as the

ONC201 target. In another recent paper also referencing the preprint, we found that Dopamine Receptor

D5 (DRD5) is a Modulator of Tumor Response to DRD2 antagonism by ONC201

<https://www.ncbi.nlm.nih.gov/pubmed/30559168>. We also cited this new paper in the newly revised

manuscript.

The same biological findings can be submitted or published in several different journals? I'm afraid that it is a duplicate publication.

> Comment 4: The authors insist that the proposed method has the following advantages over the previous methods.

> 1. Computational power – we observed a much quicker training time when using a bayesian framework compared to other machine learning approaches

> 2. Interpretability – Since predictive power is measured in terms of odds ratio and probabilities we found it was easier to understand what individual data types led to a specific prediction.

> 3. New Data – New data types could be added in without needing to retrain the entire model for the other data types.

> However, kernel-based methods in the previous works have the above features.

> For example, SITAR is computationally very efficient (*Journal of computational biology*, 18, 133-145, 2011).

>

> Response to comment 4: We do agree that there are multiple different approaches that may be used to

approach drug target prediction, however one aspect of the Bayesian approach that we believe gives it an

advantage to other approaches (including kernel based approaches) is the feature level interpretability. Often times in standard kernel based methods it is difficult to understand what exactly the method has learned or how an individual prediction is driven by different features. Since Bayesian approaches are driven by probability distributions it is much more intuitive to understand what the method is learning. Additionally since the final likelihood value is a product of the individual likelihood ratios it is easy to determine which specific similarity (structural, genomic, efficacy, etc) led to a prediction of two drugs sharing a target. We also note that unlike many computational approaches, our approach is supported by substantial experimental data on ONC201 and novel microtubule inhibitors.

I think that existing methods have such advantages. For example, SITAR is not a black-box method, so it is easy to determine which specific similarity (structural, genomic, efficacy, etc) led to a prediction.

Dear Reviewer 1,

We thank you for your response on our manuscript. We appreciate you acknowledging how our previous revisions addressed most outstanding questions. In regards to your comment about comparison with previous methods, we have prepared a response how we have addressed these concerns and have highlighted the relevant sections in the manuscript:

Comment: The authors well addressed the questions excepts the ones on the comparison with previous methods.

A clear section with results on the comparisons with all the methods (Campillos et al, wiser et al, Wang et al) using the data the authors included in BANDIT if possible. A table with the resulting performances would add robustness to the method and the paper.

Response to Comment: To address this, we added a new section in the manuscript in which we compare performance of BANDIT with 3 methods side by side, in table format, including the 2 methods suggested (Campillos et al, Science, 2008; Wang et al, PLoS Comp Biol, 2013), plus Caniza et al, 2017, Proceedings of the XLIII Latin American Computer Conference (CLEI)).

The new section starts at line 502 of the attached revised manuscript, with changes highlighted in yellow.

According to this comparison, BANDIT outperforms all 3 approaches.

A systematic comparison of performance with Keiser et al, was not feasible since they did not publish their AUC and do not have code available that would allow systematic comparison – their website only allows comparison on 1 drug at a time. However, we did compare our predictions on ONC201 and the novel microtubule inhibitors against predictions from the SEA algorithm (what was used in Keiser et al). We found that SEA did not predict DRD2 as a target of ONC201 and did not identify microtubules as a target for approximately half of the validated microtubule inhibitors we identified. These results further underline BANDIT's increased accuracy and utility.

We would like to add that our analysis of ONC201 and its experimentally validated connection to its DRD2 target has had substantial impact in that it led to repositioning of the drug in tumor types that express high levels of DRD2 such as H3K27 mutated gliomas. The drug is showing exciting signs of efficacy in this otherwise untreatable pediatric tumors. The story (which

mentions our analysis) is described in this article <https://www.philly.com/health/brain-cancer-oncoceutics-philadelphia-dopamine-fda-20181205.html>. Of all prior target prediction methods, we are not aware of any that have had such dramatic impact. As you know our paper has substantial experimental validation on other compounds on which we found the target using our method; very few if any prior target prediction methods have had that broad level of validation.

We hope you will find our revisions and clear side by side benchmarking satisfactory

Reviewer #1 (Remarks to the Author):

The authors improved the comparison with previous method section.

However, the authors should include a short description of the differences between BANDIT and the other methods. Especially, they should well compare and contrast BANDIT to these methods. Was possible to compute these methods using the six data type they used for BANDIT? If yes, do the authors could find the experimentally validated prediction they discovered via BANDIT?

Minor: For consistency, I would report the accuracy not as a percentage but as a float number.

Reviewer Comment: The authors improved the comparison with previous method section. However, the authors should include a short description of the differences between BANDIT and the other methods. Especially, they should well compare and contrast BANDIT to these methods.

Was possible to compute these methods using the six data type they used for BANDIT? If yes, do the authors could find the experimentally validated prediction they discovered via BANDIT?

Response to comment: We discussed the differences between BANDIT and these existing methods in the “Introduction” section of the manuscript when we discuss the state of the field and existing limitations. For instance:

- Lines 64-67: “However, the majority of approaches rely on structural similarity between a queried compound and a database of drugs with known targets to predict new targets for the queried compound¹⁵⁻¹⁷. Yet, by relying on only a single data type these methods are more susceptible to data-specific noise and suffer from limited utility and accuracy.”
- Lines 72-83: “However these approaches still suffer from a few limitations:
 - 1) They use known targets of a given candidate compound as an input, which limits their applicability to orphan compounds with no known targets.
 - 2) They often use gene-based similarity features, a method inherently biased against the discovery of diverse types of targets; favoring instead, the discovery of genes of the same class as the known drug-targets.
 - 3) Most models only integrate one or two additional data types in addition to compound structure.
 - 4) Many rely complex integration algorithms that are not easily able accommodate new sources of information as they become available.
 - 5) Most have only evaluated their approach on a small number of drugs (<500) without thorough experimental validation.”

The biggest difference between BANDIT and existing methods is that BANDIT is able to integrate over 6 data types (one of the methods strengths) while most existing methods are capable of only integrating 1-2 data types (see Table 1 below). If the integrated data type is something that obtained later in development (e.g. side effects, known indications, MESH terms, known binding targets), this further limits any method’s applicability and prevents its use on novel compounds still being developed (such as the ONC201 or the identified microtubule inhibitors). BANDIT’s diverse data integration capabilities not only overcome these development-stage related limitations, but also allows BANDIT to overcome biases present in an individual data type. This leads to a higher overall accuracy compared to other methods, as we have benchmarked in the paper and in Table 1. In addition to our existing discussion in the “Introduction”, we have added in several lines to restate some differences in the section: “Comparison to Existing Drug Target Prediction Methods.”

These differences also limit the possibility of direct benchmarking. Because each of these methods have been built with on a limited number of data types in mind (both in terms of the underlying methodology, but also in terms what can be used as an input for new testing), it is impossible to use BANDIT’s much larger training database as an input to some of the methods. Additionally for methods that use known side

effects or MESH terms as input to predict targets, it is impossible to obtain target predictions for ONC201 or the orphan compounds that we identified as microtubule inhibitors since these molecules do not have known side effects or MESH terms (Table 1).

For methods where it was possible to input data on ONC201 or the validated microtubule inhibitors, we continued to see better performance from BANDIT. None of the available methods were able to identify DRD2 as a predicted target for ONC201 (lines 293-295). Additionally, when we ran other structure based prediction methods (Keiser et al & Nickel et al) on the 14 validated microtubule inhibitors we found that 5 out of the 14 (36%) were not predicted by any other method besides BANDIT to target microtubules (lines 224-226).

To help with understanding these differences between existing methods and BANDIT, we have created a comparison table that can be found below (Table 1).

Table 1: Comparison of existing drug target prediction methods vs. BANDIT

Method	Method Type	Integrated Data Types	Test Set Used	Accuracy /AUC	Predicted DRD2 for ONC201
Campillos et al	Similarity network	Side Effects	DrugBank, Matador, & PDSP	 47% accuracy for top predictions (top .166% of scores) 	NA (Unable to test as ONC201 is still in development and does not have well profiled side effects)
Wang et al	Batch corrected GSEA	Transcriptional Response	DrugBank	0.66 AUC	No
Caniza et al – SIMui	Graph based GO similarity	MESH Ontology Terms	DrugBank	0.69 AUC	NA (Unable to test as ONC201 is still in development and does not have MESH Ontology terms)
Caniza et al – Resnik	Semantic similarity	MESH Ontology Terms	DrugBank	0.67 AUC	NA (Unable to test as ONC201 is still in development and does not have MESH Ontology terms)
Caniza et al – SLMgic	Graph based GO similarity	MESH Ontology Terms	DrugBank	0.65 AUC	NA (Unable to test as ONC201 is still in development and does not have MESH Ontology terms)

					terms)
Caniza et al – Lin	Semantic similarity	MESH Ontology Terms	DrugBank	0.55 AUC	NA (Unable to test as ONC201 is still in development and does not have MESH Ontology terms)
Caniza et al – Jiang and Conrath	Semantic similarity	MESH Ontology Terms	DrugBank	0.52 AUC	NA (Unable to test as ONC201 is still in development and does not have MESH Ontology terms)
Keiser et al	Ligand based similarities	Chemical Structure	MDDR, WOMBAT	NA	No
Nickel et al	Similarities based on structure and ATC codes	Chemical Structure, ATC Code	SuperTarget, ChEMBL, BindingDB	NA	No
BANDIT	Bayesian machine learning	Side Effects, Chemical Structure, Transcriptional Response, Bioassays, Growth Inhibition Screening	DrugBank	 • 0.89 AUC • 76% accuracy for top predictions (top .166% of scores) 	Yes